# Identification of a modulator of the actin cytoskeleton, mitochondria, nutrient metabolism and lifespan in yeast

Cierra N. Sing[1,2], Enrique J. Garcia[1], Thomas G. Lipkin[1,3], Thomas M. Huckaba [1,4], Catherine A. Tsang[1], Arielle C. Coughlin[1], Emily J. Yang [1], Istvan R. Boldogh[1], Ryo Higuchi-Sanabria[1,2,5] & Liza A. Pon[1,2✉]

In yeast, actin cables are F-actin bundles that are essential for cell division through their function as tracks for cargo movement from mother to daughter cell. Actin cables also affect yeast lifespan by promoting transport and inheritance of higher-functioning mitochondria to daughter cells. Here, we report that actin cable stability declines with age. Our genome-wide screen for genes that affect actin cable stability identified the open reading frame *YKL075C*. Deletion of *YKL075C* results in increases in actin cable stability and abundance, mitochondrial fitness, and replicative lifespan. Transcriptome analysis revealed a role for *YKL075C* in regulating branched-chain amino acid (BCAA) metabolism. Consistent with this, modulation of BCAA metabolism or decreasing leucine levels promotes actin cable stability and function in mitochondrial quality control. Our studies support a role for actin stability in yeast lifespan, and demonstrate that this process is controlled by BCAA and a previously uncharacterized ORF *YKL075C*, which we refer to as actin, aging and nutrient modulator protein 1 (*AAN1*).

[1] Department of Pathology and Cell Biology, Columbia University, New York, NY 10032, USA. [2] Institute of Human Nutrition, Columbia University, New York, NY 10032, USA. [3] Technology Development Group, University of California, Los Angeles, CA 90095, USA. [4] Department of Biology, Xavier University of Louisiana, New Orleans, LA 70125, USA. [5] Leonard Davis School of Gerontology, University of Southern California, Los Angeles, CA 90089, USA. ✉email: lap5@cumc.columbia.edu

The actin cytoskeleton is critical, not just for normal cellular function, but for lifespan and healthspan, the quality of life during the aging process. For example, actin function declines with age in the immune system, which contributes to defects in (1) phagocytosis by macrophages, (2) exocytosis in neutrophils, (3) formation of the immunological synapse between T cells and antigen-presenting cells, and (4) migration of mesenchymal stem cells to sites of injury[1–11]. Similarly, the speed of actin sliding on skeletal muscle myosin and maximum velocity of muscle fiber shortening decline during the aging process, which may contribute to age-linked loss of muscle function[12–16].

The actin cytoskeleton is also critical for mother-daughter age asymmetry, the process whereby babies are born young, largely independent of the age of their parents. This process is conserved from bacteria to humans and is driven at the single-cell level by the asymmetric inheritance of aging determinants, which preserve mother cell age and rejuvenate daughter cells. Oxidatively damaged, carbonylated proteins and protein aggregates are retained exclusively in yeast mother cells and defects in this process reduce lifespan and abolish mother-daughter age asymmetry[17,18]. Conversely, higher-functioning populations of organelles including mitochondria and the yeast lysosome (vacuole) are preferentially inherited by yeast daughter cells[19–22]. Interestingly, mitochondria are also asymmetrically inherited during cell division in human mammary stem-like cells, and this process affects cell fate[23].

The asymmetric inheritance of aging determinants in yeast is dependent on actin cables, tracks for polarized movement of virtually all cargoes in yeast. Actin cables assemble in the bud, extend along the mother-bud axis and disassemble in the mother cell tip[24–29]. However, in contrast to most cytoskeletal tracks, which are stationary, our previous studies revealed that these structures are dynamic and undergo retrograde flow, treadmill-like movement away from the bud and towards the mother cell[24]. As a result, actin cables act as filters to prevent the transport and inheritance of lower-functioning mitochondria from mother to daughter cells. Indeed, we found that promoting actin cable dynamics promotes mitochondrial function and extends lifespan in yeast[20].

Although cytoskeletal function is critical for lifespan control in yeast and other eukaryotes, how actin changes with age, and the mechanisms underlying this process, are not well understood. Here, we present evidence that the stability of actin cables declines with age in budding yeast. We also identify a role for a previously uncharacterized gene (YKL075C) in regulating actin cable stability, mitochondrial quality, branched-chain amino acid (BCAA) metabolism, and longevity.

## Results

**Actin cable integrity declines with age in yeast.** Here, we visualized actin cables as a function of age in yeast undergoing replicative aging, a model of lifespan control that is based on the number of times a yeast mother cell divides before senescence[30]. Briefly, mid-log phase yeast cells, which are primarily young cells, were labeled on their cell walls with biotin. These cells were then propagated in shake flasks or in a mini-chemostat aging device (mCAD), which resulted in the production of young, unlabeled cells by budding from the original biotinylated population[31,32]. Older, biotinylated cells were then separated from young, unlabeled cells using streptavidin affinity purification[31,32]. The replicative age of isolated young and old cells was determined by assessing the number of bud scars, ring-shaped structures on the cell wall of mother cells that form at the site of mother-daughter separation (Fig. 1a).

We visualized the actin cytoskeleton of young and old yeast cells by staining with fluorochrome-coupled phalloidin and imaging by super-resolution structured illumination microscopy (SIM). We confirmed that young cells contain the 2 actin-containing structures that are present throughout the cell cycle: actin patches (endosomes that have an F-actin coat and are enriched in the bud) and actin cables (bundles of F-actin that extend along the mother-bud axis and function in cargo transport) (Fig. 1a). Moreover, we observed morphological and organizational changes in the actin cytoskeleton in old yeast cells (>10 generations in replicative age). Specifically, we found that F-actin content in actin cables, assessed by the intensity of staining using fluorochrome-coupled phalloidin, decreases with age (Fig. 1a, b). Consistent with this, we found that the apparent width of actin cables declines with age (Supplementary Fig. 1a). Finally, in young cells, all actin cables are polarized along the mother-bud axis. In contrast, depolarized actin cables are evident in cells of advanced age (>10 generations) (Fig. 1a).

To determine whether the observed cytoskeletal changes are due to effects on actin stability, we assessed the sensitivity of actin cables of young and old cells to destabilization by Latrunculin A (Lat-A). Lat-A promotes loss of F-actin structures by two mechanisms. It inhibits actin polymerization by sequestering monomeric actin, and stimulates actin depolymerization by promoting release of Pi from actin bound to ADP and Pi within actin polymers[33].

Here, we used concentrations of Lat-A that preferentially affect actin cables. Previous studies indicate that high levels of Lat-A (200 μM) induces a loss of all F-actin-containing structures in yeast, while exposure to low Lat-A concentrations (1–10 μM) results in actin cable loss with no effects on actin patches (Supplementary Fig. 1b)[34]. We isolated young and old yeast using the mCAD as described above and monitored actin cable content in these cells as a function of the time of treatment with low levels of Lat-A. As expected, we observed a loss of actin cables with Lat-A treatment in young cells. In addition, we detected a statistically significant increase in the rate of Lat-A-induced actin cable loss in old compared to young yeast cells (Fig. 1c, d). Thus, actin cable stability declines with age in yeast.

**Genes that are critical for actin cable stability.** Since actin cables are essential for transport of cellular constituents to buds during bud formation and growth, Lat-A-induced loss of actin cables inhibits yeast cell growth (Fig. 1e). Therefore, to identify genes that are critical for actin cable stability, we screened a yeast strain collection that bears a deletion in each of the non-essential yeast genes for deletions that result in reduced sensitivity to Lat-A-induced growth inhibition (Fig. 1e). In all, 18 hits were identified in this screen. None of the deletions identified had effects on growth rates in the absence of Lat-A (Supplementary Fig. 1c). However, in the presence of low levels of Lat-A, all 18 deletion strains exhibited increased growth rates compared to wild-type cells (Fig. 1e and Supplementary Fig. 1d). 13 of the 18 strains also exhibited a significant increase in actin cable abundance compared to wild-type cells (Fig. 1f). Two of the hits that exhibit normal actin cable abundance in the absence of Lat-A treatment, BRO1 and FAR10, are genes that function in membrane transport. These hits, and other hits that produce the same phenotype, may impact on Lat-A sensitivity through effects on uptake or export of the drug.

To determine whether the deletions affect overall cell polarity, we studied the localization of actin patches. These structures are enriched in the bud, which reflects endocytosis at that site, presumably to promote recycling of membrane components at sites of membrane deposition during bud growth. We do not detect depolarization of actin patches in any of the hits studied. Thus, using an unbiased approach, we identified genes that, when

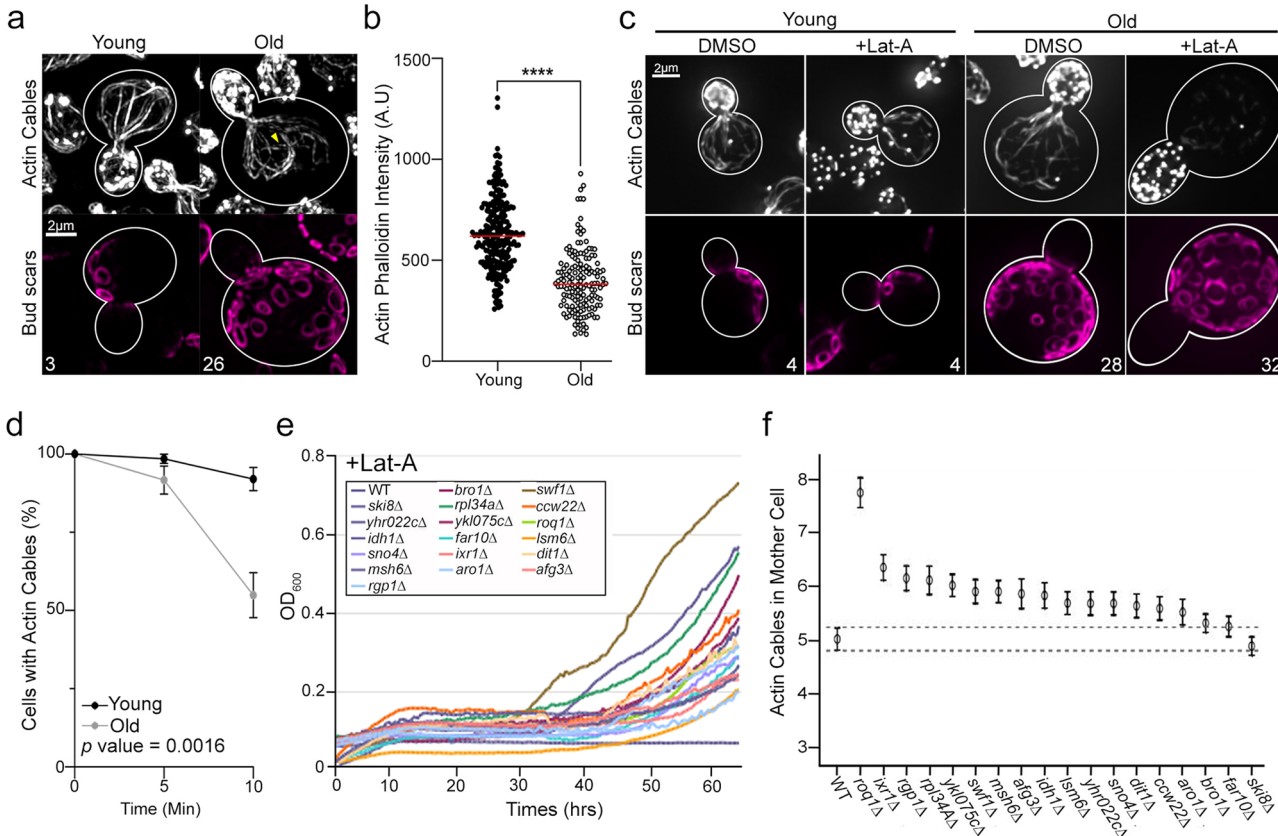

**Fig. 1 Actin stability declines with age. a** Representative SIM images of F-actin and bud scars of young and old wild-type (WT) cells stained with Alexa488-phalloidin and Alexa594-wheat germ agglutinin (WGA594), respectively. Yellow arrow identifies a depolarized actin cable. **b** Quantification of Alexa488-phalloidin corrected mean signal intensity in mother cells of young and old cells isolated from mCAD. Combined $n$ of cells/strain from 3 trials: 248 (young) and 152 (old). The $p$ value ($< 1.00e^{-15}$) was determined by a two-tailed non-parametric Mann–Whitney test. **c** Representative SIM images of Alexa488-phalloidin-stained actin and Alexa594-WGA-stained bud scars in young and old WT cells treated with vehicle (DMSO) or 1 μM Lat-A for 10 mins. **d** Time-course of actin cable loss during treatment with 1 μM Lat-A. Greater than 126 cells/timepoint/condition/trial were analyzed for $n$ of 3 trials. Error bars: SEM. **e** Growth curves of deletion strains that exhibit reduced sensitivity to Lat-A (10 μM) in our genome-wide screen. **f** Actin cable abundance in WT cells and the 13 hits from the screen. Actin was visualized by rhodamine-phalloidin staining and wide-field fluorescence microscopy. Error bars: SEM. Scale bars, 2 μm.

deleted, increase actin cable abundance and reduce yeast cell sensitivity to destabilization of actin cables by Lat-A treatment.

Two of the hits identified in the screen encode previously uncharacterized open reading frames, *YHR022C* and *YKL075C*. Moreover, none of the hits were in genes that encode known actin binding or regulatory proteins. Therefore, we carried out SGD GO SLIM analysis of the 17 characterized gene hits. This analysis revealed a role for two or more of the hits in processes including response to chemical stressors, cell cycle, DNA recombination, regulation or repair, amino acid metabolism, translation, protein targeting, and endosomal transport (Supplementary Fig. 1e).

**YKL075C affects actin cable stability and yeast lifespan.** Our efforts focused on *YKL075C*, one of the two uncharacterized genes identified in our screen. Since strains in the yeast deletion collection can carry secondary mutations with functional consequences[35], we constructed new *YKL075C* deletion strains. In newly constructed strains, we confirmed that deletion of *YKL075C* (1) does not affect yeast cell growth or polarity (Fig. 2a, b), (2) reduces the sensitivity of yeast to the growth-inhibiting effects of low-level Lat-A treatment (Fig. 2a), and (3) increases actin cable abundance (Fig. 2b, c). Since another open reading frame (*YKL076C*) overlaps with the C-terminal coding region of *YKL075C*, it is possible that the phenotypes observed in *ykl075cΔ* cells are due to loss of the overlapping gene. However, we find that expression of the *YKL075C* gene under

control of its endogenous promoter in *ykl075cΔ* cells reduced the actin cable abundance to wild-type levels (Supplementary Fig. 2a, b). Thus, we found that ectopic expression of *YKL075C* complements loss of the gene. Together, our findings support a role for *YKL075C* in modulating actin cable abundance in budding yeast.

Next, we tested whether deletion of *YKL075C* promotes actin cable stability using the Lat-A sensitivity assay described above. We found that actin cables in *ykl075cΔ* cells exhibit reduced sensitivity to Lat-A treatment compared to wild-type yeast (Fig. 2d, e). We also found that deletion of *YKL075C* has no effect on the rate of retrograde actin cable flow (Supplementary Fig. 2c, d). Thus, we identified a modulator that affects actin cable stability and abundance but has no detectable effect on actin cable treadmilling.

Since actin cable stability declines with age, we tested whether stabilization of actin cables by deletion of *YKL075C* promotes yeast cell lifespan. Here, we studied the replicative lifespan (RLS) of yeast, the number of times that a mother cell divides before senescence. We found that deletion of *YKL075C* extends lifespan in yeast. Specifically, the mean RLS of *ykl075cΔ* cells (26 generations) is significantly greater than that of wild-type cells (19 generations) (Fig. 2f). Although reduced RLS can result from nonspecific factors, interventions that extend RLS must affect processes that limit lifespan. Thus, our finding that deletion of *YKL075C* extends RLS supports a role for that gene in the aging process in yeast.

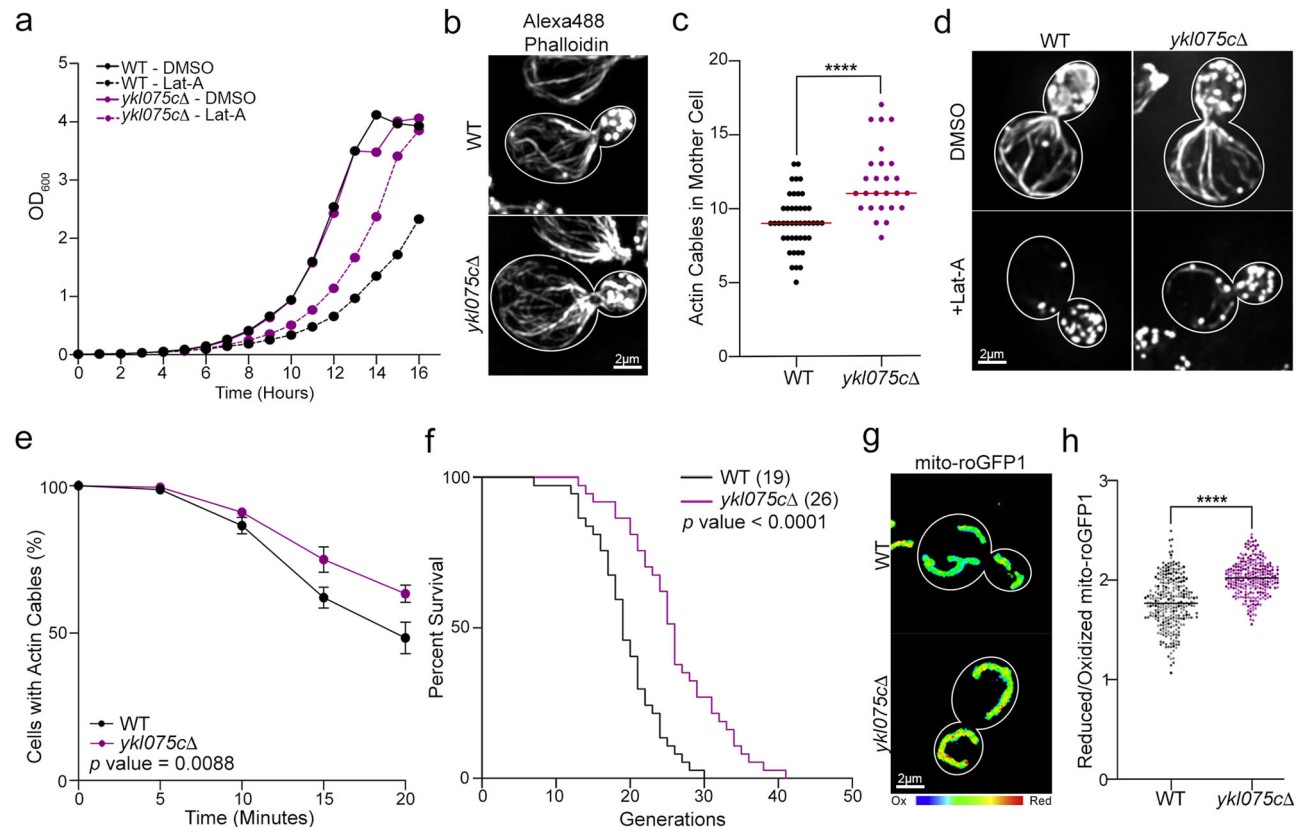

**Fig. 2 Identification of a gene (*YKL075C*) that affects actin cable stability, mitochondria, and lifespan. a** Growth curves of WT and *ykl075cΔ* cells treated with vehicle (DMSO) or 2 μM Lat-A for 16 h. Data from a representative trial is shown (*n* = 3 trials). **b** Representative images of Alexa488-phalloidin stained F-actin of WT and *ykl075cΔ* cells. **c** Actin cable abundance in mother cells of mid-log phase WT and *ykl075cΔ* cells. *n* = 44 cells (WT) and 26 cells (*ykl075cΔ*). Data from a representative trial is shown (*n* = 3 trials). *p* value (1.00e$^{-06}$) was determined by a two-tailed non-parametric Mann-Whitney test. **d** Representative images of Alexa488-phalloidin stained F-actin of WT and *ykl075cΔ* cells treated with vehicle (DMSO) (top panels) or 2 μM Lat-A for 20 min (bottom panels). **e** Quantitation of actin cables in WT and *ykl075cΔ* cells during treatment with 2 μM Lat-A. Greater than 256 cells/ strain/timepoint were analyzed for *n* = 5 trials. Error bars: SEM. *p* value (8.8e$^{-03}$) was determined by simple linear regression analysis. **f** RLS of WT and *ykl075cΔ* cells determined by manual micromanipulation. *n* of cells/strain: 35 (WT) and 36 (*ykl075cΔ*). *p* value (1.23e$^{-06}$) was determined by the two-sided Mantel-Cox test. Data from a representative trial from *n* of 3 trials. **g** Representative images of mitochondrial redox state (reduced:oxidized ratio) visualized using mdito-roGFP1 in WT and *ykl075cΔ* cells. Higher numbers/warmer colors indicate more reducing mitochondria. **h** Quantification of mito-roGFP1 redox ratios in WT and *ykl075cΔ* cells. Combined *n* of cells/strain from three trials: 284 (WT) and 260 (*ykl075cΔ*). *p* value (< 1.00e$^{-15}$) was determined by a two-tailed non-parametric Mann–Whitney test. Scale bars, 2 μm.

Finally, we assessed the role of *YKL075C* in cellular healthspan by examining two hallmarks of cellular fitness: (1) mitochondrial function and (2) mean generation time (the time required for a cell to complete a round of cell division). Mitochondrial function was evaluated using a ratiometric sensor for the redox state of the mitochondrial matrix (mito-roGFP1)[19,36,37]. We found that mitochondria are more reduced, and therefore higher functioning, in *ykl075cΔ* cells compared to wild-type cells (Fig. 2g, h). Equally important, we found deletion of *YKL075C* does not affect asymmetric inheritance of mitochondria in yeast: fitter mitochondria, that are more reduced, are preferentially inherited by yeast daughter cells in both *ykl075cΔ* and wild-type cells (Supplementary Fig. 2e). Thus, deletion of *YKL075C* promotes the function of mitochondria, an aging determinant that is asymmetrically inherited in yeast.

Consistent with this, we found that the mean generation time (MGT) is reduced in *ykl075cΔ* cells (Supplementary Fig. 2f). We confirmed previous findings that the MGT of wild-type yeast cells increases with age. We also found that the MGT of young (0-10 generation) wild-type and *ykl075cΔ* cells is similar. This finding provides additional evidence that deletion of *YKL075C* does not affect the growth rate of mid-log phase yeast, which are primarily

young cells. While the MGT of *ykl075cΔ* cells increased with age, the rate of increase of MGT with age in *ykl075cΔ* cells is significantly less than that of wild-type cells. Thus, we identified a role for the previously uncharacterized protein, Ykl075cp, in modulating actin cable stability and abundance, lifespan and two markers of healthspan, mitochondrial function, and generation time.

### *YKL075C* targets branched-chain amino acid homeostasis.
*YKL075C* has no obvious functional domains or sequence similarity to known proteins. Since *YKL075C* affects actin cables, we tested whether the protein encoded by the *YKL075C* gene (Ykl075cp) co-localizes with the actin cytoskeleton in yeast. We tagged the endogenous *YKL075C* gene at its C terminus with GFP and found that the signal obtained by fluorescence imaging was weak. This finding is consistent with reports that Ykl075cp is not an abundant protein[38]. Therefore, we tagged the gene with 13 copies of the Myc epitope. We found that tagging *YKL075C* at its chromosomal locus results in expression of a protein of the expected apparent molecular weight (Supplementary Fig. 3a), and does not disrupt the cell growth rate. Consistent this, we found

that expression of Myc-tagged *YKL075C* complements loss of the gene and restores actin cable abundance to wild-type levels (Supplementary Fig. 3b, c). Since tagging *YKL075C* with 13-Myc does not affect its function, we studied the localization of the tagged protein and actin cytoskeleton in mid-log phase yeast (Fig. 3a). Ykl075cp localizes to punctate cytosolic structures in mother cells and buds. There is no obvious co-localization of Ykl075cp with actin patches or cables. Thus, *YLK075C* effects on actin cables is not through direct effects on those structures or on endosomes.

Next, we used RNA-Seq to study gene expression in yeast bearing a deletion in *YKL075C* (Supplementary Table 1). This analysis revealed transcriptional changes in processes ranging from amino acid metabolism and the mitochondrial electron transport chain to iron homeostasis and transport (Fig. 3b). Among these, there were significant decreases in levels of transcripts encoding branched-chain amino acid (BCAA) biosynthetic processes in *ykl075cΔ* compared to wild-type cells (Fig. 3b, c). Notably, *BAT1*, which encodes the BCAA transaminase (BAT) that catalyzes the last step of BCAA synthesis[39], and 5 out of 10 other BCAA biosynthetic genes are down-regulated in *ykl075cΔ* cells. In addition, there is an increase in expression of *BAT2*, a *BAT1* paralog that preferentially mediates BCAA catabolism[40,41].

Interestingly, GO term analysis revealed that other hits identified the Lat-A screen function in amino acid metabolism or endosomal transport, a process that impacts on the amino transport and on the organelle (the vacuole) that stores and contributes to amino acid sensing. These findings support a role for amino acids in modulation of the actin cytoskeleton and raise the possibility that other hits identified in the screen may function with Ykl075cp in cytoskeletal control pathways.

To validate the transcriptome studies, we assessed the levels of specific BCAA biosynthetic genes and of BCAAs. Using qPCR, we observed a 65% reduction in *BAT1* expression, and a 2-fold increase in *BAT2* expression in *ykl075cΔ* cells compared to wild-type cells (Fig. 3d, e). We also detected a significant reduction in the level of free BCAAs in whole-cell extracts from *ykl075cΔ*, *bat1Δ*, and *ykl075cΔ bat1Δ* cells compared to wild-type cells (Fig. 3f). Importantly, while we observed a decrease in BCAA levels in both single and double deletion strains compared to wild-type cells, we found no additive effect of the simultaneous deletion of *YKL075C* and *BAT1* compared to *bat1Δ* single mutants. These findings indicate that (1) *YKL075C*, like *BAT1*, modulates BCAA levels, and (2) *YKL075C* and *BAT1* function in the same pathway for BCAA control. Collectively, our findings indicate that *YKL075C* affects actin cables, key nutrients, and lifespan in yeast. Therefore, we will refer to *YKL075C* as *AAN1*: actin, aging and nutrient modulator protein 1.

TORC1 (Target of rapamycin complex 1) regulates anabolic cellular processes and lifespan in response to nutrients including BCAAs[42–44]. Therefore, TORC1 may contribute to Aan1p function in modulating actin cables and lifespan. To test this hypothesis, we tested whether TORC1 is activated by deletion of *AAN1*. *TOR1* and *TOR2*, the 2 TOR paralog genes of yeast, are subunits of TORC1[42,45]. Since *TOR2* is an essential gene, *TOR1* nulls were used as control to evaluate TORC1 activity[46]. We confirmed previous findings that deletion of *TOR1* delays re-entry into the cell cycle after release from rapamycin-induced inhibition of cell growth[47,48]. Specifically, we found that the growth of *tor1Δ* cells is reduced compared to wild-type cells after removal of rapamycin from the growth medium (Fig. 3g, h). On the other hand, the recovery of *aan1Δ* cells from rapamycin treatment is indistinguishable from that of wild-type cells and distinct from that observed in *tor1Δ* or *aan1Δ tor1Δ* mutants (Fig. 3g, h).

To confirm these results, we examined the effect of deletion of *AAN1* on phosphorylation of ribosomal protein S6 (Rps6p), a direct downstream target of TORC1. Since there are two *TOR* genes in yeast, deletion of *TOR1* reduces Rps6p phosphorylation, while treatment with rapamycin fully inhibits TORC1-mediated Rps6p phosphorylation[49,50]. We confirmed that deletion of *TOR1* inhibits but does not block Rps6p phosphorylation. Equally important, we find that phosphorylation of Rps6p is indistinguishable in *aan1Δ* and wild-type cells (Fig. 3i, j). Consistent with this, our RNA-Seq studies indicate the gene expression pattern in *aan1Δ* cells is distinct from that observed upon TORC1 inhibition[51–57]. Collectively, these results indicate that Aan1p function is not dependent upon TORC1.

**YKL075C targets BCAA metabolism through effects on actin**. Next, we tested whether modulating BCAA levels can drive a change in cytoskeletal stability in wild-type or *aan1Δ* cells. Here, modulation of BCAAs was conducted in prototrophic yeast strains, which can synthesize BCAAs. Moreover, amino acid depletion or supplementation was carried out for short time periods (20–30 min). Thus, the experimental conditions were designed to alter BCAA levels without eliciting starvation or stress responses.

We found that deletion of *BAT2* has no obvious effect on actin cables (Supplementary Fig. 4a, b). On the other hand, *bat1Δ* mutants exhibit an increase in actin cable abundance similar to that observed in *aan1Δ* cells (Fig. 4a, b). In addition, we found that short-term (20 min) depletion of leucine from the growth medium does not produce cytoskeletal phenotypes (e.g. global destabilization of the actin cytoskeleton) associated with acute stress. Instead, it results in an increase in actin cable abundance in wild-type cells, reaching the elevated levels observed in *aan1Δ* cells (Fig. 4c, d). Moreover, we found that short-term depletion of another amino acid, lysine, has no detectable effect on actin cable abundance (Supplementary Fig. 4c, d). Conversely, leucine supplementation reduced the amount of F-actin in actin cables, as assessed by F-actin staining intensity, in both wild-type and *aan1Δ* cells, and resulted in reduced actin cable length in *aan1Δ* cells (Supplementary Fig. 4e–g). It also resulted depolarization of the actin cytoskeleton as assessed by the distribution of actin cables and patches (Supplementary Fig. 4g, h). Finally, we found that deletion of *BAT1* or depletion of leucine does not affect actin cable abundance in *aan1Δ* cells (Fig. 4a–d). Collectively, these findings support a role for BCAA in modulation of actin cables. They also indicate that Aan1p and BCAA function in the same pathway for control of actin cable abundance.

Next, we used the Lat-A sensitivity assay described above to determine whether deletion of *BAT1* and the associated decrease in BCAA levels affect actin cable stability. We found that actin cable stability is greater in *bat1Δ* cells compared to wild-type cells and similar to that observed in *aan1Δ* and *aan1Δ bat1Δ* cells (Fig. 4e, f). Overall, these findings indicate that (1) increased actin cable abundance observed in *bat1Δ*, *aan1Δ*, and *aan1Δ bat1Δ* cells may be due to effects on actin cable stability, and (2) *BAT1* and *AAN1* function in the same pathway for control of actin cable stability.

Finally, we found that deletion of *BAT1* and *AAN1* also produces similar effects on mitochondrial function. Assessing mitochondrial redox state using mito-roGFP1, we found that mitochondria in *bat1Δ* and *aan1Δ* cells are more reduced and therefore higher functioning compared to mitochondria in wild-type cells. Moreover, we found that the mitochondrial redox state is similar in *bat1Δ* and *aan1Δ* cells (Fig. 4g, h). Finally, higher-functioning mitochondria are preferentially inherited by daughter cells in *bat1Δ* mutants, as in *aan1Δ* and wild-type cells (Supplementary Fig. 4i). These findings indicate that *BAT1* and *AAN1* have similar effects on mitochondrial function: deletion of either gene promotes mitochondrial function but has no effect on asymmetric inheritance of the organelle.

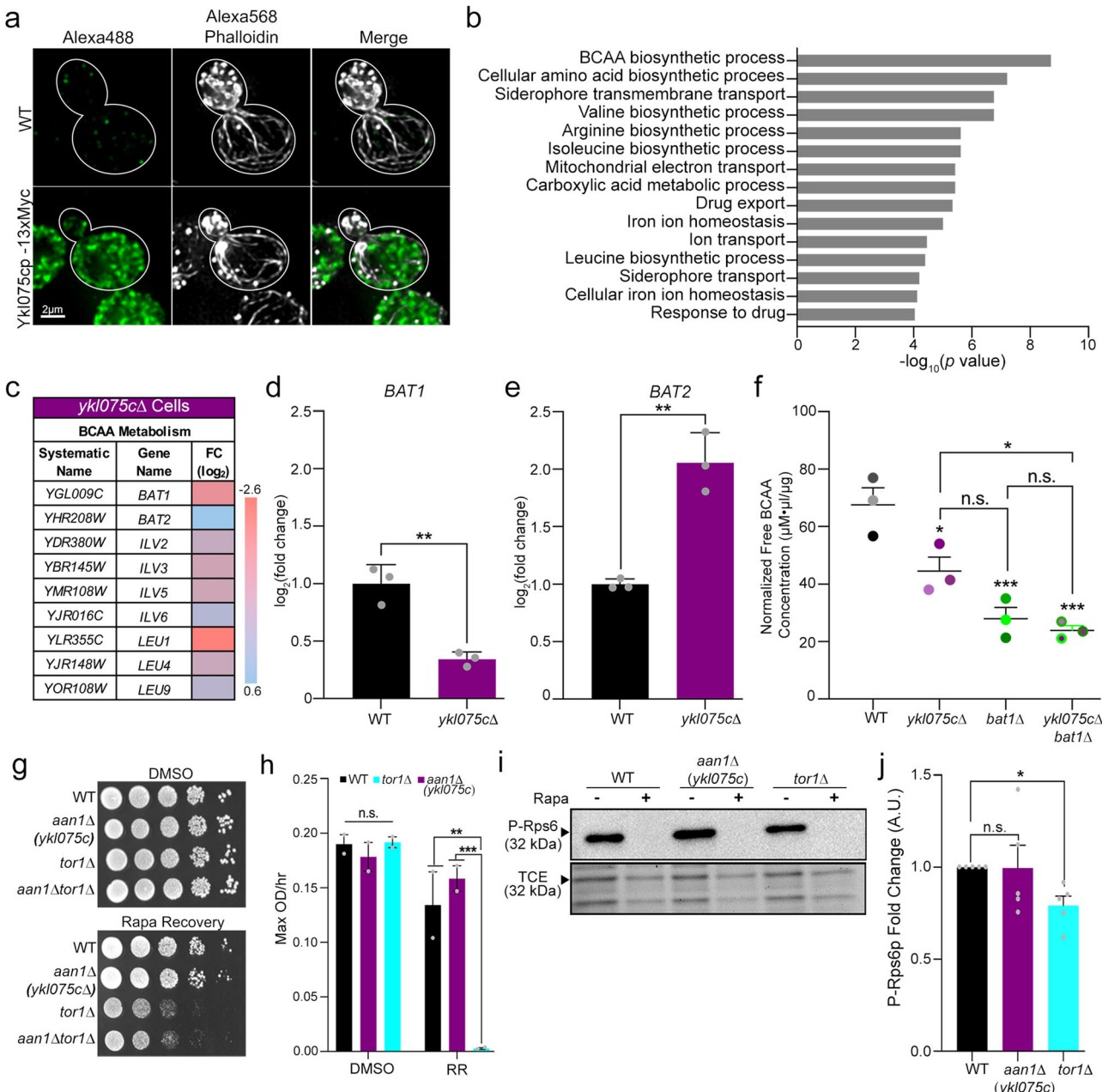

**Fig. 3 Ykl075cp protein localization and links to branched-chain amino acid (BCAA) metabolism. a** Localization of Ykl075cp and the actin cytoskeleton in cells containing either native *YKL075C* or *YKL075C* tagged with the Myc epitope of 3 trials. Ykl075cp-13Myc visualized by immunofluorescence using anti-Myc antibodies and Alexa488-tagged secondary antibodies. The actin cytoskeleton was visualized by staining with Alexa568-phalloidin. Scale bar, 2 μm. **b** Top 15 gene ontology (GO) terms of *ykl075cΔ* cells. Two-side multiple hypothetic correction test with Bonferroni correction statistical analysis from Yeastract+ database. **c** Table of BCAA proteins with altered transcript levels in *ykl075cΔ* cells. **d, e** Validation of mRNA transcript levels of BCAA transaminases *BAT1* and *BAT2* by quantitative PCR. Decreased *BAT1* and increased *BAT2* gene expression in *ykl075cΔ* cells are significantly different from WT cells (unpaired two-tailed t-test; *p* values: 2.9e$^{-03}$ (WT vs. *bat1Δ*) and 3.3e$^{-03}$ (WT vs. *bat2Δ*)). Data is representative of three trials. **f** Quantification of free intracellular BCAA levels in WT, *ykl075cΔ*, *bat1Δ*, and *ykl075cΔ bat1Δ* cells using Cell BioLabs BCAA colorimetric ELISA kit. *p* values: 2.43e$^{-03}$ (WT vs. *ykl075cΔ*), 9.00e$^{-04}$ (WT vs. *bat1Δ*), 5.00e$^{-04}$ (WT vs. *ykl075cΔ bat1Δ*), 1.50e$^{-01}$ (*ykl075cΔ* vs. *bat1Δ*), 4.16e$^{-02}$ (*ykl075cΔ* vs. *ykl075cΔ bat1Δ*), and 9.10e$^{-01}$ (*bat1Δ* vs. *ykl075cΔ bat1Δ*). **g** Serial dilutions of WT, *aan1(ykl075c)Δ*, *tor1Δ*, and *aan1Δ tor1Δ* cells recovering from either DMSO or 200 nM rapamycin treatment on YPD plates. **h** Growth rate (OD$_{600}$/h for 72 h) of WT, *aan1(ykl075c)Δ*, *tor1Δ*, and *aan1Δ tor1Δ* cells during recovery from DMSO or 200 nM rapamycin treatment (RR). *p* values of two biological repeats: 8.49e$^{-01}$ (WT$_{DMSO}$ vs. *aan1Δ*$_{DMSO}$), 9.96e$^{-01}$ (WT$_{DMSO}$ vs *tor1Δ*$_{DMSO}$), 8.06e$^{-01}$ (*aan1Δ*$_{DMSO}$ vs *tor1Δ*$_{DMSO}$), 5.21e$^{-01}$ (WT$_{RR}$ vs. *aan1Δ*$_{RR}$), 1.90e$^{-03}$ (WT$_{RR}$ vs *tor1Δ*$_{RR}$), and 8.00e$^{-04}$ (*aan1Δ*$_{RR}$ vs *tor1Δ*$_{RR}$). **i** Representative western blot of the phosphorylation of ribosomal protein-S6 (Rps6p) in mid-log phase WT, *aan1(ykl075c)Δ*, and *tor1Δ* cells in the presence or absence of rapamycin (200 μM). TCE total protein loading control. **j** Quantification of Rps6p phosphorylation of panel i of untreated or rapamycin-treated WT, *aan1(ykl075c)Δ*, and *tor1Δ* cells. Data are representative of three trials. *p* value: 3.50e$^{-02}$ (non-parametric Kruskal-Wallis test). Error bars: SEM for panels **d-f**, **h** and **j**.

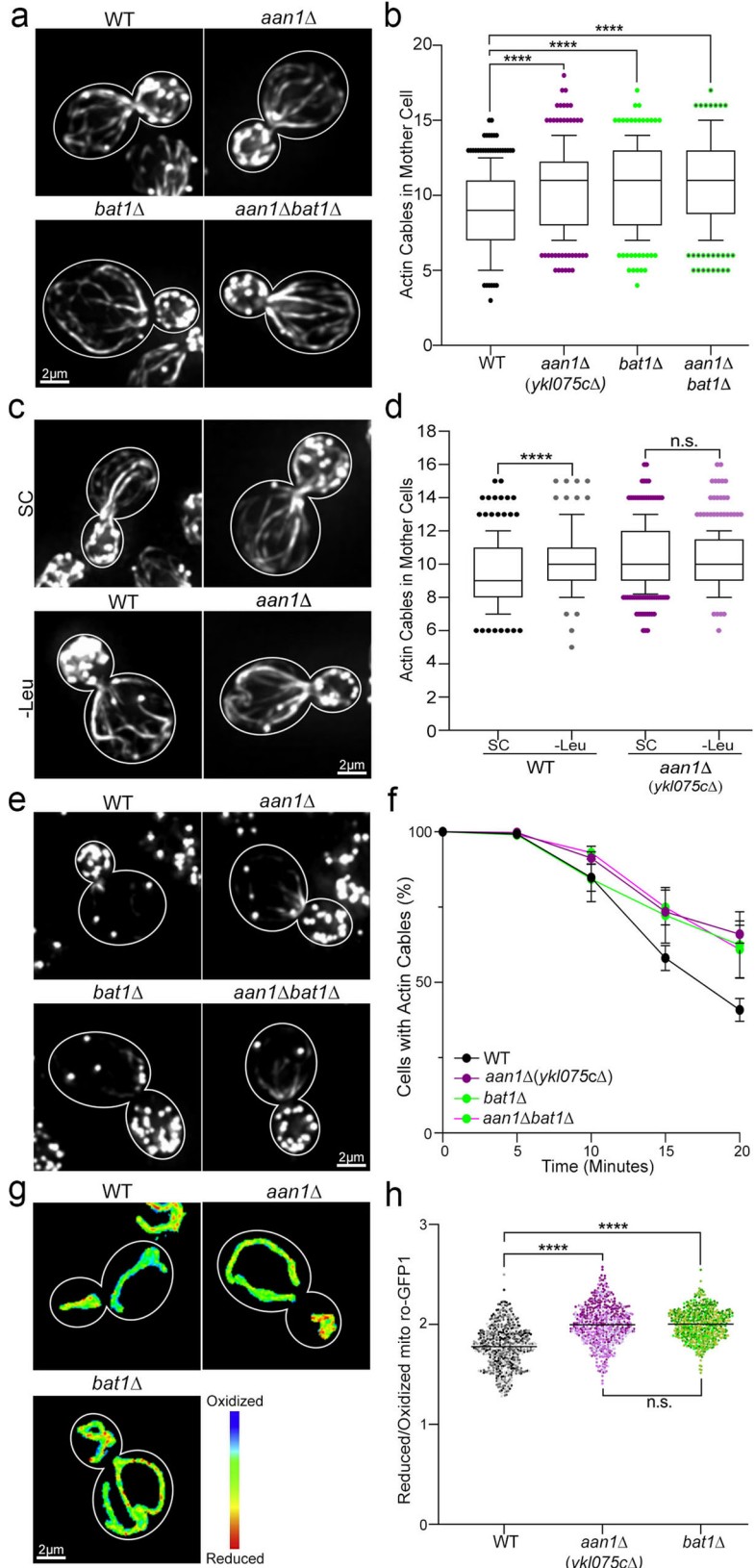

## Discussion

Age-associated declines in actin cytoskeletal function occur in mammalian cells, tissues, and organs. However, the mechanism underlying this process is not well understood. Here, we report that the stability of actin cables, essential components of the yeast actin cytoskeleton, declines with age. We identified a previously undescribed open reading frame, which we refer to as *AAN1*, that impacts on actin cable stability, mitochondrial function, BCAA metabolism, and cellular lifespan. Moreover, we obtained evidence that modulation of BCAA metabolism or levels of leucine

**Fig. 4 AAN1(YKL075C) affects actin cables and mitochondria through its function in BCAA homeostasis. a** F-actin visualized using Alexa488-phalloidin in mid-log phase WT, aan1Δ, bat1Δ, and aan1Δ bat1Δ cells. **b** Quantitation of actin cable abundance in panel **a**. The center band of the box and whisker plot represents the median; the box indicates middle quartiles; whiskers extend to the 10th and 90th percentiles. Combined n of cells/strain from 3 trials: 274 (WT), 246 (aan1Δ), 236 (bat1Δ), and 258 (aan1Δ bat1Δ). P values: $5.16e^{-08}$ (WT vs. aan1Δ), $9.50e^{-10}$ (WT vs. bat1Δ), $5.40e^{-11}$ (WT vs aan1Δ bat1Δ), >0.999 (aan1Δ vs bat1Δ), >0.999 (aan1Δ vs aan1Δ bat1Δ), and >0.999 (bat1Δ vs. aan1Δ bat1Δ). P values were generated from two-tailed non-parametric Kruskal–Wallis tests. **c** AlexaFluor488-phalloidin stained F-actin of mid-log phase WT, aan1Δ, bat1Δ or aan1Δ bat1Δ cells grown in synthetic complete (SC) medium in the presence or absence of leucine (+/−Leu). **d** Box and whisker plot of actin cable abundance +/−Leu. The center band represents the median; the box indicates middle quartiles; whiskers extend to the $10^{th}$ and $90^{th}$ percentiles. Combined n of cells/strain/group: 242 (WT$_{SC}$), 216 (WT$_{-Leu}$), 331 (aan1Δ$_{SC}$), and 313 (aan1Δ$_{-Leu}$). p values were generated from two-tailed non-parametric Mann–Whitney test of 3 trials: $5.88e^{-09}$ (WT$_{SC}$ vs. WT$_{-Leu}$) and $2.08e^{-01}$ (aan1Δ$_{SC}$ vs. aan1Δ$_{-Leu}$). **e** Alexa488- phalloidin stained F-actin of mid-log phase WT, aan1Δ, bat1Δ, and aan1Δ bat1Δ cells treated with 2 μM Lat-A for 20 min. **f** Actin cable loss as a function of the time of Lat-A treatment in WT, aan1Δ, bat1Δ, and aan1Δ bat1Δ cells. Error bars: SEM of three trials (n = 100 cells/strain/trial). p values: $1.30e^{-03}$ (WT vs. aan1Δ), $3.63e^{-02}$ (WT vs. bat1Δ), $1.99e^{-02}$ (WT vs. aan1Δ bat1Δ), $7.45e^{-01}$ (aan1Δ vs. bat1Δ), $6.83e^{-01}$ (aan1Δ vs. aan1Δ bat1Δ), and $9.87e^{-01}$ (bat1Δ vs. aan1Δ bat1Δ). p values were determined by simple linear regression analysis. **g** Mitochondrial redox state visualized using mito-roGFP1. **h** Reduced:oxidized mito-roGFP1 ratios in WT, aan1Δ, bat1Δ, and aan1Δ bat1Δ cells. Combined n of cells/strain of three trials: 546 (WT), 591 (aan1Δ), 581 (bat1Δ). p values: $<1.00e^{-15}$ (WT vs aan1Δ), $<1.00e^{-15}$ (WT vs. bat1Δ), and >0.999 (aan1Δ vs. bat1Δ). p values were determined using a two-tailed non-parametric Kruskal–Wallis test. Scale bars, 2 μm.

affects actin cable stability and abundance. Finally, we found that *AAN1* function is not dependent upon the TORC1 pathway for sensing nutrients and regulating lifespan.

**The stability of the actin cytoskeleton declines with age in yeast.** Our studies focused on actin cables, essential components of the yeast actin cytoskeleton that serve as tracks and movement generators for organelles and other cargoes during asymmetric yeast cell division. Actin cables assemble at sites of polarized cell surface growth in the bud by direct interactions with the polarisome, a protein complex that stimulates actin polymerization for actin cable assembly[58,59]. Polarisome-dependent insertion of newly polymerized actin into the end of actin cables drives elongation of actin cables and promotes extension of those structures from the bud into the mother cell.

We detected age-linked declines in the amount of actin polymer within actin cables. Consistent with this, we observed an increase in the sensitivity of actin cables to destabilization by Lat-A in old yeast (>10 generations in replicative aging) compared to young yeast. These findings indicate that actin cable stability declines with age in yeast. We also detect defects in actin cable organization in yeast as they undergo replicative aging. Specifically, we detect depolarized actin cables, which do not align along the mother-bud axis, in cells of advanced age. This defect in actin cable polarization may reflect age-linked alterations in cell polarity. Alternatively, they may reflect fragmentation of destabilized actin cables, and release of those fragments from their interactions with the polarisome and cell polarity machinery of yeast. These models are not mutually exclusive.

Recent studies in *Caenorhabditis elegans* revealed age-associated declines in the integrity of the actin cytoskeleton in muscle, intestine and hypodermis, and a role for PAT-10, a calcium-binding actin-associated protein, in that process[60,61]. Interestingly, deleting *pat-10* destabilizes the actin cytoskeleton and results in premature aging; while *pat-10* overexpression has the opposite effect[61]. Since essential elements of the yeast actin cytoskeleton also undergo a decrease in stability with age, our findings indicate that actin cytoskeletal destabilization may be a conserved aging phenotype in eukaryotic cells.

**Identification of a regulator of actin cables, mitochondria, and lifespan in yeast.** We carried out a screen of a library of 4848 yeast strains, which each bear a deletion of a non-essential yeast gene, for strains that exhibit reduced sensitivity to Lat-A. The concentration of Lat-A used for the screen destabilizes actin cables and inhibits yeast cell growth, but does not affect other actin-containing structures. Actin cable components and

assembly factors were not identified as hits in this screen. Deletion of these proteins destabilizes actin cables and is expected to increase Lat-A sensitivity. Instead, the screen revealed 18 gene deletions that promote yeast growth in the presence of low levels of Lat-A. 13 of these deletion strains also exhibit an increase in actin cable abundance in the absence of Lat-A. GO term analysis revealed that hits that impact on Lat-A sensitivity and actin cable abundance function in amino acid metabolism and endocytosis. This raises the possibility that other hits identified in our screen may contribute to Aan1p function in modulation of the actin cytoskeleton in response to BCAA.

Our efforts focused on the previously uncharacterized open reading frame *YKL075C*, which we refer to as *AAN1*. Deletion of *AAN1* results in an increase in actin cable abundance. It also results in reduced sensitivity of actin cables to destabilization by treatment with Lat-A. Finally, we found that deletion of *AAN1* results in extended replicative lifespan and increased healthspan as assessed by improved mitochondrial redox state and reduced mean generation time during cell growth. Thus, we obtained evidence for a role for *AAN1* in actin cable stability and abundance, mitochondrial quality control and replicative lifespan in yeast.

Our previous studies revealed a role for actin cables in control of lifespan through effects on mitochondrial quality control. Specifically, we found that promoting actin cable dynamics results in inheritance of higher-functioning mitochondria by yeast daughter cells, which promotes cell fitness and lifespan[20]. Although deletion of *AAN1* promotes mitochondrial function, extends replicative lifespan and affects actin cable stability, there was no effect on the rate of retrograde actin cable flow. Thus, the increase in mitochondrial function and lifespan observed in *aan1Δ* cells is not due to effects on actin cable dynamics.

Moreover, although deletion of *AAN1* promotes mitochondrial function, it is possible that other actin cable-dependent aging determinants also contribute to *AAN1* effects on lifespan. For example, defects in the polarisome or destabilization of the actin cytoskeleton results in premature aging in yeast, by inhibiting retention of protein aggregates, an aging determinant, in the mother cell[17,18]. Actin cables also serve as tracks for inheritance of vacuoles in yeast[62] and inheritance of more acidic, higher-functioning vacuoles has also been linked to mitochondrial function and lifespan control in yeast[22]. Thus, *AAN1* effects on actin cables may impact on lifespan through effects on mitochondria and other aging determinants.

Finally, previous studies also support a role for the actin bundling protein, Scp1p, in mitochondrial quality control, actin dynamics, and lifespan control[63]. Since Scp1p localizes to actin patches, it affects actin dynamics on structures other than actin cables. Moreover, Scp1p affects lifespan in stationary phase, not

dividing yeast cells, whereas Aan1p affects lifespan in dividing yeast cells. Thus, available evidence indicates that Aan1p and Scp1p affect lifespan by different cytoskeleton-based mechanisms. These observations underscore the complexity of the actin cytoskeleton and its impact on mitochondria and lifespan.

**BCAAs are regulated by Aan1p and control actin cable abundance and stability**. Aan1p has no conserved protein domains or no homology to any known protein. It localizes to punctate cytoplasmic structures that do not colocalize with actin cables or actin patches. Analysis of the transcription signature of *AAN1* deletion cells revealed an unexpected link between *AAN1* and BCAA metabolism: deleting *AAN1* decreases the expression of BCAA biosynthetic genes, and increases the expression of a BCAA degradation gene. Moreover, we find decreasing BCAA biosynthesis by deleting *BAT1* or depleting a specific BCAA, leucine, in wild-type cells produces an increase in actin cable abundance and stability that resembles that observed in *aan1Δ* cells. Finally, reducing BCAA, by deletion of a BCAA biosynthetic enzyme or leucine depletion, did not affect actin cable abundance in *aan1Δ* cells.

An excess of BCAAs is toxic and is the basis for Maple Syrup Urine Disease, an inherited metabolic disorder that is characterized by neuronal dysfunction and cerebral atrophy[64]. Previous studies indicate that leucine supplementation results in actin-dependent changes in the morphology of astrocytes[64]. Similarly, in yeast, we find that short-term supplementation with leucine results in destabilization of actin cables and changes in actin cable morphology, in wild-type and *aan1Δ* cells. We also provide the evidence that depletion of leucine can promote actin cable abundance and stability and that Aan1p function in control of actin cables is dependent on its function in regulating BCAAs. However, leucine supplementation is required for chronological lifespan extension in non-dividing yeast cells[65,66]. Thus, leucine effects on lifespan control are complex and may be influenced by dose and context.

The mechanism of action of Aan1p remains to be determined. Aan1p does not localize to actin cables, mitochondria or the nucleus. Therefore, it has indirect effects on the actin cables, mitochondrial function, and BCAA gene expression. Moreover, we find that Aan1p function is not dependent upon TORC1: deletion of *AAN1* does not affect the phosphorylation of a conserved TORC1 substrate or recovery from rapamycin. It also does not produce a pattern of gene expression observed upon TORC1 activation. However, Aan1p may affect the General Amino Acid Control pathway (GAAC), a signaling pathway that regulates yeast cell growth, metabolism, translation, and lifespan in response to amino acid limitations. Indeed, GAAC affects expression of amino acid biosynthesis genes, including *BAT1*, and aminoacyl-tRNA synthetase[67]. Moreover, the actin cytoskeleton has been linked to GAAC function in translational regulation[68–71]. Ongoing studies focus on the mechanisms underlying *AAN1* control of BCAA metabolism, and how BCAAs control the actin cytoskeleton and lifespan in yeast. Finally, uncovering Aan1p establishes a link between energy metabolism and the actin cytoskeleton, which may impact on health and disease, including the metabolism and migration of cancer cells.

## Methods

**Yeast growth conditions**. All *S. cerevisiae* strains used in this study are derivatives of the wild-type BY4741 strain (*MATa his3Δ1 leu2Δ0 met15Δ0 ura3Δ0*) or the S288C strain (*MATα SUC2 gal2 mal2 mel flo1 hap1 ho bio1 bio6*) from Open Biosystems (Huntsville AL). All cultures used for imaging and RLS studies were grown to mid-log phase ($OD_{600}$ 0.1–0.5) by shaking at 30 °C in a glucose-based yeast-peptone-dextrose (YPD) or synthetic complete (SC) medium.

**Yeast strain construction**. All knockout strains (Supplementary Table 2) were created using homologous recombination-based methods. Genes of interest were

replaced with either auxotrophic or drug resistance markers using primers and cassettes described in Supplementary Table 3. For example, *ykl075cΔ* cells were generated by replacing *YKL075C* with KanMX6, a geneticin resistant gene, using a homologous recombinant PCR fragment, which contained: flanking sequences of the *YKL075C* gene and the KanMX6 coding region from the plasmid pFA6-KanMX6 (Addgene, Watertown MA). All transformations were carried out using the lithium acetate and transformants were identified as colonies that grow on selective solid media. For example, KanMX6-positive cells were selected using YPD solid media containing 200 μg/mL Geneticin (Sigma-Aldrich, St. Louis MO). Selected colonies were further validated using PCR sequencing.

The same techniques were used to add fluorescent protein tags at the chromosomal loci of genes of interest using PCR fragments amplified from plasmids: pFA6a-GFP(S65ST)-KanMX6 and pFA6a-GFPEnvy-KanMX6 (Addgene, Watertown MA). Selection for positive transformants was carried out as described above and confirmed by visual analysis using wide-field fluorescence microscopy.

**Column isolation of aged yeast cells**. Cells of defined replicative age were isolated using a modification of the protocol described in Smeal et al.[31]. Yeast strains of interest were grown to a mid-log phase at 30°C in YPD media ($OD_{600}$ < 0.5). In all, $10^7$ cells were removed from the mid-log phase culture, concentrated by centrifugation using a Sorvall ST-16 tabletop centrifuge (3500 RPM for 4 min at RT), and washed twice with 1x PBS at pH 8.0. The cell pellet was resuspended with 500 μL of sterile 1x PBS. To biotinylate cells, 3.0 mg of EZ Link Sulfo-NHS-LC-Biotin (Thermo Fischer Scientific, Waltham MA) was dissolved in 500 μL of 1x PBS at room temperature (RT) and immediately added to cells. Cells were incubated for 30 min on a rotating platform at RT. Biotinylated cells were gently washed twice with 1x PBS 100 mM Glycine, inoculated into a sterile flask containing 500 mL of YPD media and propagated for 8 h (6–8 generations) at 30°C with shaking. After propagation, cells were incubated with FluidMAG-Streptavidin beads (Chemicell, Berlin Germany) at 500 μg/5.0 × $10^7$ cells for 30 min at RT. Cells were then washed with YPD and applied to an LS column (Miltenyi Biotec Inc., Auburn CA) mounted in a QaudroMACS magnetic separator rack (Miltenyi Biotec Inc., Auburn CA). Labeled (old) cells were retained in the column, and unlabeled cells in the column flow-through were recovered and used as young cell controls. The column was washed with YPD until the flow-through did not contain any cells. To isolate old cells, the LS column was removed from the magnetic separator rack and the column was washed with YPD media. Isolated of old cells were further aged by propagation as described above for 8 h. After 8 h, old cells were isolated as described above.

**mCAD isolation of aged yeast cells**. Aged-yeast cells were isolated from the miniature-Chemostat Aging Device (mCAD) as described by Hendrickson et al.[32]. Briefly, 4 × $10^6$ cells were obtained from an overnight mid-log culture with a cell density of <0.2 $OD_{600}$ and washed twice with 1x PBS containing 0.25% PEG3350. The cell pellet from washed cells was resuspended in 1x PBS, mixed with 2 mg of EZ Link Sulfo-NHS-LC-Biotin (Thermo Fischer Scientific, Waltham MA) dissolved in 1x PBS, and incubate for 30 min at RT in a rotating platform mixer. Biotinylated cells were washed with 1x PBS 0.25% PEG3350, inoculated into a sterile flask with 500 mL of SC media and propagated for 8 h at 30 °C with shaking to mid-log growth state ($OD_{600}$ < 0.3). Dynabeads MyOne Streptavidin C1 beads (Thermo Fischer Scientific, Waltham MA) were washed twice with SC media, and resuspended in 20 mL of SC in a 50 mL conical tube. Biotinylated-labeled cells were incubated with Dynabeads (9 μg/$10^6$ cells) for 15 min at RT in rotating platform mixer. The cell mixture was transferred into a sterile glass culture tube surrounded by ring magnets, and allowed to stand and for 5 min at RT. Unlabeled young cells that were not immobilized in the magnetic field were removed by aspiration. Ring magnets were then removed and cells that were released were suspended in 1 mL of SC media and transferred to a Luer lock syringe for mCAD loading.

The mCAD was prepared and assembled as previously described[32]. Labeled cells in the Luer lock syringe were loaded onto the Luer needle entry port of the mCAD vessel. The cell loaded-vessel was placed into ring magnets for 10–15 min to immobilize the labeled cells before they were propagated in the mCAD vessel for 41–45 h with constant media feed using a peristaltic pump (25 mL/h) and aeration using an air pump (0.8–1.0 PSI). To collect cells, all entry ports were removed from the mCAD vessel. Unbound cells were carefully removed by a serological pipette and discarded and residual unbound cells were removed by washes with media. The vessel was removed from the magnetic rings and released cell were resuspend in SC media, washed with fresh media and resuspended in SC media.

**Growth rates**. Growth curves were measured using an automated plate reader (Tecan; Infinite M200, Research Triangle Park NC) or manually using spectrophotometer (Beckman; Irvine CA). Strain preparation for automated plate reader detection follows as previously described in Garcia et al.[72]. Each strain is grown to mid–log phase in rich, glucose-based media (YPD) and diluted to an $OD_{600}$ of 0.07 (2.0 × $10^6$ cells/mL). In all, 10 μL of the diluted strain was added to a well containing 200 μL YPD in a 96-well plate. Cells were propagated at 30 °C without shaking, and an optical density reading at 600 nm (OD600) was measured every 20 min for 72 h. Alternatively, growth rates can be measured manually by a spectrophotometer. An overnight mid-log culture ($OD_{600}$ 0.3–0.5) in YPD was prepared and diluted to 0.1 $OD_{600}$. Cells were propagated in a 30 °C shaking

incubator and optical density reading at 600 nm was measured in spectro-photometer cuvette at a 1 in 10 dilution every hour for 16 h.

**Microscopy**. Wide-field fluorescence imaging was carried out using an Axioskop 2 microscope equipped with a 100x/1.4 Plan-Apochromat objective (Zeiss, Thornwood NY), an Orca-ER CCD camera (Hamamatsu Corporation, Bridgewater NJ), a pE-4000 LED illumination system (coolLED, Andover UK). The system was controlled by NIS Elements 4.60 Lambda software (Nikon). Super-resolution imaging was performed using a structured illumination microscope (N-SIM S, Nikon) equipped with a 100x/1.49 oil-immersion objective lens (Nikon Instruments, Melville NY), an EMCCD Camera (iXon, Andor Technology Ltd, Belfast Ireland) and NIS Elements software (Nikon Instruments, Melville NY).

**Visualization of F-actin cytoskeleton and bud scars**. Mid-log phase yeast cells were concentrated to a cell density of 0.5 $OD_{600}$ and resuspended in 1 mL of growth medium for fixed-cell staining. Cells were fixed in 3.7% paraformaldehyde at 30 °C with shaking for 50 min.

We visualized the actin cytoskeleton in fixed cells by wide-field microscopy using the following protocol. Fixed-cells were washed three times with 1x PBS, followed by one wash with 1x PBT (PBS containing 1% w/v BSA, 0.1% v/v Triton X-100, 0.1% w/v sodium azide), and actin was stained with 2.5 μM AlexaFluor488-phalloidin (Thermo Fischer Scientific, Waltham MA) for 25 min at RT in the dark. Cells were subsequently washed three times with 1xPBS and resuspended in 7–10 μL of SlowFade Diamond anti-fade mountant (Thermo Fischer Scientific, Waltham MA). 1.8 μL of stained cells prepped in mountant solution were placed on a glass slide and covered with a #1.5 coverslip. AlexaFluor488-phalloidin was excited with a 470 nm LED and an ET470/40x filter (Chroma, Bellow Falls VT). Z-stacks were captured at 0.3 μm intervals using 1 × 1 binning, 200–300 ms exposure time, and gain of 116. Using Volocity 5.5 (Quorum Technologies Ltd, Lewes UK), images were deconvolved using a constrained iterative restoration algorithm assuming 507 nm emission wavelength with a 100% confidence limit and 60 iterations. All images were contrast-enhanced with similar parameters.

Cells visualized by SIM were fixed by paraformaldehyde as previously described. Bud scars were stained first with 5 μg/mL AlexaFluor594-WGA (Thermo Fischer Scientific, Waltham MA) and incubated at RT in the dark for 30 min. Cells were washed twice with 1x PBS and once with 1x PBT. After bud scar staining, the actin cytoskeleton was stained with 2.5 μM AlexaFluor488-phalloidin staining as previously described. Additional preparations were required for SIM imaging and used the following protocol described in Higuchi-Sanabria et al.[73]. Briefly, stained cells adhered to coverslips coated with concanavalin-A were mounted on a slide with ProLong Diamond anti-fade mountant (Thermo Fischer Scientific, Waltham MA). Prepared slides cured overnight in the dark at RT. Used N-SIM S instrument as described above with a camera EM gain of 200. AlexaFluor488-phalloidin was excited with a 488 nm laser at 45% power with 300 ms exposure time and AlexaFluor594-WGA was excited with a 561 nm laser at 10% power with 200 ms exposure time. A z-series of 0.125 μm intervals with 1×1 binning was collected through the entire cell. SIM images were reconstructed using parameters described in Higuchi-Sanabria et al.[73]. All images were contrast-enhanced with similar parameters.

**Immunofluorescence staining**. Mid-log phase yeast cells (0.2–0.5 $OD_{600}$ cell density) were fixed with 3.7% paraformaldehyde for 1 h at 30 °C with shaking. Immunofluorescence staining was performed as described in detail by Higuchi-Sanabria et al.[73]. Fixed cells were treated with 50 μg/mL of Zymolyase 20 T (Seikagaku Inc., Tokyo Japan) for 30 min to remove the yeast cell wall. Spheroplasts (10 μL) were resuspended in 100 μL 1x PBS and placed onto a polylysine-coated 22×22 mm coverslip, and incubated in a dark humid chamber for 35 min at RT. Unbound spheroplasts were gently washed away with 1x PBT, and the spheroplast-coated coverslip was placed onto 30 μL of 1:100 diluted primary 13xMyc antibody (Developmental Studies Hybridoma Bank, University of Iowa, Iowa City IA) and incubated for 2 h in a dark humid chamber at RT. Coverslip was rinsed four times with 1x PBT and incubated in 30 μL of 1:100 diluted secondary antibody (Goat anti-Mouse IgG Alexa Fluor plus 488, Ref A32723, Thermo Fischer Scientific, Waltham MA) for 1 h. Coverslip was rinsed with 1x PBT and mounted on a slide with 1.5 μL of SlowFade mountant (Thermo Fischer Scientific, Waltham MA). The coverslip edges were sealed with clear nail polish and slides were stored at 4 °C.

**Actin cable stability and abundance analysis**. For cell growth curves, yeast cells were incubated with 8 μM Latrunculin-A (Lat-A, Sigma-Aldrich, St. Louis MO) for 3 days, taking cell density measurements ($OD_{600}$) every hour. Yeast cells were treated with 1-2 μM Lat-A for 20 min and 5 min interval samplings were acquired, fixed, and stained to determine actin cable stability. Stained cells were imaged with wide-field microscopy as described above. Quantified deconvolved wide-field images for the presence or absence of actin cables in mother cells of both DMSO and Lat-A treated yeast cells. The binary quantification determined the rate of actin cable loss during the Lat-A time-course.

Actin cable abundance was quantified using deconvolved wide-field images. Actin cables were mapped from the bud neck to the mother cell tip using the Volocity tracing tool and the number of actin cable tracings determined the actin cable abundance.

**Analysis of mitochondrial redox state using mito-roGFP1**. A ratiometric probe, mito-GFP1, was used to measure the redox state of the mitochondrion. We performed imaging and analysis as previously described in McFaline-Figueroa et al.[19] and Vevea et al.[37]. Ratiometric probe, mito-roGFP1, was integrated at the *HO* locus. Acquired images on a wide-field microscope with reduced and oxidized states excited at 470 nm and 365 nm with a GFP filter cube without an excitation filter (Zeiss filter 46HE, dichroic FT 515, emission 535 nm/30). Images were captured images with a z-series at 0.3 μm intervals through the entire depth of the cell. The imaging deconvolution parameters were 60 iterations of a constrained iterative restoration algorithm with a calculated PSF using the following parameters: $\lambda_{ex}$ 507 nm and 100% confidence limit (Volocity 5.5, Quorum Technologies Ltd, Lewes UK). Once the background was subtracted and thresholded each image, the reduced/oxidized ratio calculated by dividing the intensity of the reduced channel ($\lambda_{ex}$ = 470 nm, $\lambda_{em}$ = 525 nm) over the intensity in the oxidized channel ($\lambda_{ex}$ = 365 nm, $\lambda_{em}$ = 525 nm), in Volocity software.

**RLS analysis**. RLS measurements were performed as previously described, without alpha-factor synchronization[30]. Selected strains stored at −80 °C were streaked onto YPD plates and grown for 2 days at 30 °C. A swab from a colony patch was grown overnight in liquid YPD at 30 °C to mid-log phase. In all, 10 μL of cell suspension was streaked onto a YPD plate. Small-budded cells were isolated and arranged in a matrix using a micromanipulator mounted on a dissecting microscope (Sporeplay, Singer Instruments, Somerset UK). Upon a complete budding event, mother cells were removed and discarded to leave only virgin cells. The time and number of subsequent daughter cells produced by each virgin mother cell were recorded until all replication ceased.

**Branched-chain amino acid levels**. Cell lysate was prepared from 8 to 10 $OD_{600}$ of mid-log phase cells by vortexing cells with 0.5 mm glass beads in chilled 1x PBS for 5 min. Beads and debris were pelleted from the cell suspension at 13 RPM for 0.5 s and the supernatant was placed on ice. Total protein concentration was measured using a Pierce BCA protein assay kit (Thermo Fischer Scientific, Waltham MA). BCAA levels were determined using the Branched Chain Amino Acid Assay kit (Cell Biolabs Inc., San Diego CA) following the manufacturer's instructions. BCAA levels were measured in triple technical replicates against negative control samples lacking leucine dehydrogenase. Levels of free BCAAs extracted were normalized to the extracted protein concentration from cell lysates and intracellular free BCAA levels were calculated based on the leucine standards.

**Western blot analysis**. Cell lysates were prepared from mid-log phase yeast cells (0.5 $OD_{600}$) as previously described Liao et al.[74]. Protein lysates were resuspended in 150 μL of 1xSDS sample buffer containing protease inhibitor cocktail, vortexed with 100 μL of 0.5 mm glass in 4 °C for 5 min, and incubated at 100 °C for 5 min. Protein lysates were loaded onto a SDS-PAGE gel containing 0.5% trichloroethanol (TCE). TCE was crosslinked to protein by exposing the gel to UV light at 300 nm for 2.5 min after the completion of electrophoresis[75] and cross-linked proteins were detected using ChemiDoc MP Imaging System (Bio-Rad, Hercules CA). TCE-crosslinked proteins were used as protein loading control. SDS-PAGE gels were transferred to a PVDF membrane (Immobilon-FL, EMD Millipore, Billerica MA). The PVDF membrane was either incubated in blocking solution (5% skim milk or 5% BSA in 1xPBS) for 1 h, followed incubation with by primary and secondary antibodies. Proteins were detected using Luminata Forte Western HRP substrate (MiliporeSigma, Burlington MA) and ChemiDoc MP Imaging System. The primary antibodies used in these studies were: 1:5000 dilution of rabbit polyclonal antibodies against P-RPS6 at Ser235/236 (#2211, Cell Signaling Technology, Danvers MA), and 1:1000 dilution of mouse monoclonal antibodies against 13xMyc tag (Developmental Studies Hybridoma Bank, University of Iowa, Iowa City IA).

**RNA sequencing**. The transcriptome was analyzed as previously described in Liao et al.[74]. RNA was extracted from mid–log phase WT and *ykl075cΔ* cells using the RNeasy kit (Qiagen, Germantown, MD). RNA quality was analyzed with Agilent 2100 Bioanalyzer using a Plant RNA Nano chip and RNA Integrity Number (RIN) scores higher than nine were sequenced. RNA library preparations and sequencing reactions were conducted at GENEWIZ, LLC. RNA sequencing libraries were prepared using the NEB Next Ultra RNA Library Prep Kit for Illumina using manufacturer's instructions (NEB, Ipswich MA). The sequencing library was validated on the Agilent Tape Station (Agilent Technologies, Santa Clara CA), and quantified using a Qubit 2.0 Fluorometer (Invitrogen, Thermo Fischer Scientific, Waltham MA) as well as by quantitative PCR (KAPA Biosystems, Sigma-aldrich, St. Louis MA). The sequencing libraries were clustered on a single lane of a flowcell. After clustering, the flowcell was loaded on the Illumina HiSeq instrument 4000 according to the manufacturer's instructions. The samples were sequenced using a 2 × 150 bp Paired End (PE) configuration. Sequence reads were trimmed to remove possible adapter sequences and nucleotides with poor quality using Trimmomatic v.0.36. The trimmed reads were mapped to the *Saccharomyces cerevisiae* S288C reference genome available on ENSEMBL using the STAR aligner v.2.5.2b. Unique gene hit counts were calculated by using feature counts from the Subread package v.1.5.2. After extraction of gene hit counts, the gene hit counts table was used for downstream differential expression analysis. Using DESeq2, a comparison of gene

expression between WT and *ykl075cΔ* cells was performed. A two-tailed Wald test was used to generate *p*-values and log₂ fold changes. Genes with a *p*-value <0.05 and absolute log₂ fold change >1 were called as differentially expressed genes. Yeastract+ database was to determine GO terms from RNA seq[76].

**cDNA synthesis and quantitative PCR**. RNA was extracted from cultured mid-log phase WT or *ykl075cΔ* cells using the RNeasy kit (QIAGEN, Germantown, MD). Genomic DNA contamination was removed using TURBO DNA-free Kit (Ambion, Thermo Fischer Scientific, Waltham MA). In all, 1 mg of DNA-free RNA was used for cDNA synthesis with SuperScript IV First-Strand Synthesis System (Invitrogen, Thermo Fischer Scientific, Waltham MA). cDNA was diluted and used for quantitative PCR reaction with PowerUp SYBR Green Master Mix (Applied Biosystems, Foster City CA). Primers for qPCR were designed using NCBI Primer Blast (https://www.ncbi.nlm.nih.gov/tools/primer-blast/) with a PCR product size of 100 bp and max Tm difference of 2 °C. For each specified gene, ΔCT was calculated as $CT_{gene} - CT_{actin}$, and fold change was calculated as $2^{-\Delta\Delta CT}$ with actin serving as the endogenous control for each sample.

**Quantification and statistical Analysis**. GraphPad Prism 9 (GraphPad Software, San Diego CA) was used to conduct all statistical analyses and create all graphs. All tests applied two-tailed statistical analysis. *p* values for simple two-group comparison were determined with a two-tailed Student's *t* test for parametric distributions and a Mann–Whitney test for non-parametric data. For multiple group comparisons, *p* values were determined by a 1-way ANOVA Kruskal–Wallis test with Dunn's post hoc test for non-parametric distributions. Survival curves were statistically compared using a Log-rank (Mantel-Cox) test. Bar graphs show the mean and SEM; in box and whisker graphs, the box represents the middle quartile, the midline represents the median, whiskers show the 10th and 90th percentiles, and points indicate outliers. For all tests, *p* values are classified as follows: \**p* < 0.05; \*\**p* < 0.01; \*\*\* *p* < 0.001; \*\*\*\**p* < 0.0001, unless exact *p* values are noted in the figure legends.

**Reporting summary**. Further information on research design is available in the Nature Research Reporting Summary linked to this article.

## Data availability

All data supporting findings in this study are available within the paper, Supplementary Information, and source data files. Primers were designed using *S. cerevisiae* reference genome from *S. cerevisiae* genome database (https://www.yeastgenome.org/) and are available in the provided source data file. This studies RNA-seq data is publicly available on NCBI GEO database under accession code GSE174157. Gene ontology analysis was determined from Yeastract+ database (http://yeastract.com/).

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

## Acknowledgements
We thank the members of the Pon laboratory for their technical assistances and valuable discussion. This work was supported by grants from the National Institutes of Health (NIH) (GM122589 and AG051047) to L.A.P., (AG055326) to CNS, and (AG65200) to R.H.-S. N-SIM microscope used in this study are supported in part through the NIH/NCI grant (CA013696 and OD014584).

## Author contributions
This study was built on the findings from screens generated by T.G.L. and T.M.H. R.H.-S. initially conceived and conducted initial experiments for this project. C.N.S. further conceived this project, conducted experiments with major contributions from E.J.G., C.A.T., A.C.C., and E.J.Y., and wrote the manuscript. L.A.P. conceived, directed the project, and wrote the manuscript. I.R.B. edited the manuscript and provided guidance on experimental design and interpretation. All authors discussed and interpreted the data together.

## Competing interests
All authors declare no competing interests.
