## [Peer Review File · Nature Communications]

Identification of a novel modulator of the actin cytoskeleton, mitochondria, nutrient metabolism and lifespan in yeastREVIEWER COMMENTS

Reviewer #1 (Remarks to the Author):

This paper reports on the role of actin cable stability on yeast replicative lifespan and the effect of an unknown open reading frame, YKL075C, thereupon. Deletion of the YKL075C ORF stabilized actin cables, enhanced mitochondrial function (or the reduced state of mitochondria) and prolonged lifespan through its effect on the transcript of genes involved in branched-chain amino acid (BCAA) metabolism (depletion of leucine could mimic the effect of removing YKL075C).

Although the exact biochemical function of YKL075C is not deciphered, the effects of removing the ORF is very interesting and links AA metabolism with actin cable function and lifespan 'control'. The paper is very well written and much of the core data intriguing. There are a few remaining issues, which the authors might want to consider and I have listed those below.

1. Since some of the strains in the yeast deletion collection are known to carry secondary mutations, the authors constructed new YKL075C deletion strains to verify that the phenotypes observed were linked to the ORF. However, this is not a sufficient protocol to test this as the deletion could have polar effects on both downstream and upstream genes. This is particularly important here because another ORF (YKL076C) overlaps with the C-terminal region of YKL075C. A standard complementation test should be performed (providing the ORF in trans) to make sure the phenotypes are linked to the YKL075C ORF.

2. The YKL075C ORF was identified here by virtue of the strain carrying this deletion being less sensitive to Lat-A. Did the authors test if this could be due also to effects on Lat-A import or export. This seems relevant to test since the phenotypes linked previously to a YKL075C deletion includes effects on 'drug export' and 'responses to drugs'. I do believe the authors have good reasons to conclude that the ORF affects actin cables (in a Lat-A independent manner) since they observe effects without Lat-A but may want to comment on the possibility that the ORF also affects Lat-A uptake/export.

A similar argument could be applied to the differential effects of Lat-A in young and old cells – e.g. are old cells displaying defects in e.g. drug export?

3. It is somewhat difficult to follow the authors' conclusions concerning Tor: One of the conclusions states that the results indicate that Aan1p function is TORC1-independent. However, what is measured is the recovery of mutant cells

after the removal of rapamycin from the medium and the authors found that cells completely devoid of Tor1 recovers slowly (as expected) whereas *aan1Δ* cells did not. While this shows that *aan1Δ* cells do not show the same defect as TOR1-deleted cells, it does not necessarily demonstrate that the effects of deleting AAN1 is completely independent on TORC1. For example, does the *aan1Δ* have the same effect in mutants with constitutive Tor activity?

Also, the authors state that 'we find that TORC1 activity is not affected by deletion of AAN1'. This is not measured in the paper. To do so, the authors could use the standard methods of measuring RPS6 phosphorylation or a shift of HA-tagged Sch9. More experiments or a rephrasing of the text might be needed here.

4. To check the localization of the ORF protein, the authors tagged the gene with 13 copies of the Myc epitope and, using immunofluorescence, found that Ykl075cp localized to punctate cytoplasmic structures that do not co-localize with actin cables. They further show that the tag had no effects on the cell growth rate. However, to test if it is functional, this analysis needs to be complemented with analysis of whether cells carrying the tagged version, when being the sole source of Ykl075cp in the cell, is not displaying any of the phenotypes displayed by cells harboring the deleted ORF. Also, it is difficult to ascertain that the tag is not causing effects on localization and foci formation without further controls. I think the authors might want to consider removing this data set in the paper as it does not contribute much to the otherwise interesting story and raises several questions.

5. As it has been shown previously that increasing actin dynamics, either by specific conditional

actin alleles or by removing the actin-bundling protein Scp1p, increases lifespan, accompanied by affects on the polarization of the mitochondrial membrane and ROS production (Gourlay et al. JBC, 2004), it would be interesting to hear the authors view on whether Aan1 and Scp1 may act in the same pathway of lifespan control and whether deleting these two genes are additive or not.

Comment: There is some data in the Rizzolo et al., paper (2017, Cell Reports) that might be helpful to the further analysis by the authors of YKL075C as the ORF has been linked, by genetic/physical interaction studies to the 'Chaperone Cellular Network', and especially linked to the GO function 'Chromatin and Nucleic acid'. (Figure 3 and Figure S6 in the paper)

Reviewer #2 (Remarks to the Author):

This is a great paper where the authors identify a novel gene target for regulation of actin cable stability in yeast, and demonstrate that deletion of this regulator actin cable stability is increased for ageing cells and this increases viability of these cells. The authors identify that the new gene target may be linked to branch chain amino acid (BCAA) metabolism, and find that a similar effect can be found by either deleting a biosynthetic gene associated with BCAA biosynthesis or by reducing leucine levels in the medium.

The paper presents some very solid results and the findings are surely of sufficient general interest to merit publication in Nature Com, but the story needs a little more additional work in order to be sufficiently conclusive.

The authors basically find that deletion of AAN1 (the new target gene identified) gives the same effect as deleting of BAT1, that is engaged in BCAA biosynthesis. They also found that BCAA levels in both strains decreases and this points to a role of BCAA, which is further confirmed by an effect of eliminating leucine from the medium. The authors do not see any effect on TORC1. This opens up for the following questions that should be addressed:

- 1) Could Aan1p be working downstream of TORC1? Could maybe be tested by phospo-proteomics of Aan1p with our without leucine in the medium.
- 2) Is there a link to energy metabolism? BCAA can serve as an important energy source, and BCAA catabolism plays an important role in some cancer cells.
- 3) Could the authors express AAN1 heterologously to evaluate if it a DNA binding protein , i.e. is it a transcription factor.

Maybe not all experiments are needed, but some more hints towards a possible function of Aan1p would significantly increase the value of the paper.

Reviewer #3 (Remarks to the Author):

Sing et al. report that the stability of actin cables apparently declines with age in *Saccharomyces cerevisiae*. By screening a yeast deletion library, the authors identified a previously uncharacterized gene (YKL075C) as being an important factor required for proper actin cable stability and abundance, mitochondrial function and branched-chain amino acid (BCAA) metabolism. Based on the phenotypic analysis, they name YKL075C AAN1 (actin, aging and nutrient regulatory protein 1). AAN1 is not an essential gene, but strains carrying null alleles exhibit enhanced mitochondrial function, altered leucine catabolism and increased replicative lifespan.

Overall, the experimental execution and phenotypic analyses are competently done, and

manuscript is very well written. However, the major conclusions are not placed in context with previous studies showing the importance of leucine in longevity studies. Furthermore, the results from the screen leading to the identification of AAN1 are merely cursorily discussed. The inactivation of ANN1 was one of 18 null mutations that resulted in resistance to low levels of latrunculin A, and one of 15 that correlated with increased actin cable stability. The lack of discussion regarding the other null mutations makes the focus and interpretation on Ann1 function harder to understand, the overall context is missing. The authors applied transcriptomic analysis to understand Ann1 function, which led to the insight that leucine metabolism has a link to actin cable stability. Although, this is an important step forward, in the end, the manuscript does not provide a framework to place this finding in any mechanistic perspective. Consequently, the manuscript provides descriptive information that represents novel information of a little studied gene, but ultimately provides limited novel insight to advance the understanding of the coupling between BCAA metabolism, actin cable stability and longevity.

The authors may want to consider the following points:

Major comments:

1. to place the results regarding leucine in perspective and to strengthen their conclusion the authors could acknowledge previous work including: Aris et al. (2013) Autophagy and leucine promote chronological longevity and respiration proficiency during calorie restriction in yeast. *Experimental Gerontology* 48:10, 1107-1119; Maruyama et al (2016) Availability of Amino Acids Extends Chronological Lifespan by Suppressing Hyper-Acidification of the Environment in *Saccharomyces cerevisiae* <https://doi.org/10.1371/journal.pone.0151894>
2. the latrunculin A screen proved to be very informative. A brief discussion regarding the other genes identified in the screen would add significantly, perhaps organized with potential function. For example, Far10 and Bro1 did not affect number of actin cables but exhibited resistance to Lat-A. Far10 is a paralog of VPS64, Bro1 is involved in protein turnover – Rsp5 – MVB pathway. Several other genes may affect the MVB pathway, including Rgp1, Swf1, Roq1 (YJL144W - why not use gene name?). Could this provide clues that BCAA uptake is affected? Also, several of the genes have known mitochondrial function, e.g., SNO4, IXR1, IDH1 and AFG1. How does the deletion of these genes square with the idea that deletion of ANN1 leads to increased mitochondrial function? Also, the finding that ARO1 is potentially interesting, or? Finally, YHR022C is an ORF of unknown function, why focus on YKL075C?
3. It is difficult to assess what strains were used in the figures. What data stems from leu2 auxotrophs and what data is derived from LEU2 prototrophs?
4. The punctate localization of the myc-tagged Ann1 is intriguing. Does Ann1 co-localize with Golgi or endosomal markers?
5. decreases BCAA biosynthesis appears to be a key feature, and the simultaneous enhanced expression of BCAA catabolism is interesting. The increased levels of BAT2 observed in ann1 null mutants did not come up in the transcriptomics analysis, is there a reason for this?
6. the authors write that Ann1 regulates actin cables, key nutrients, and lifespan in yeast. The term regulate is a strong word – the data merely show that AAN1 is required for proper actin cable thickness, BCAA metabolism and cell longevity – where is the data to substantiate regulation?
7. SD-leu represents a very intense leucine starvation conditions for leu2 strains. Is there any indication of a stress response may increase/stabilize actin cables? Added leucine suppresses the actin stabilisation, which suggests that leucine limitation is important.
8. Does D-leucine suppress the latrunculin A affect on actin cables? The thought being that leucine may have an allosteric role in this activity.

Minor comments:

1. Healthspan? Quality of life during aging? Is this an accepted term for yeast? How to assess "quality" of life in yeast?

2. Fig 1e – the color scheme used makes it difficult to follow the growth of the strains. These curves appear to be generated using microtiter plate readers with short pathlengths for OD measurements – hence the low ODs of saturated cultures. It would be good to state in the methods how the curves were generated. By contrast, the OD measurements in Fig. 2a end up at higher ODs, likely to have determined using standard spectroscopic measurements using a 1 cm pathlength cuvette. Also, the growth curves are clearly biphasic, the mutants start to grow and then cease, seemingly to recover after 30 hr. What is going on here? Do the authors have thoughts regarding this?

3. Fig. 1e,f - in Fig.1f Lsm6 is used but in Fig. 1e. ism6 is used as strain label – ism6 does not exist in SGD, probably a mistake in the label in Fig. 1e?

Response to Reviewer Concerns

The Reviewers comments are shown in bold font and our responses are show in plain font.

Reviewer #1 (Remarks to the Author):

1. Since some of the strains in the yeast deletion collection are known to carry secondary mutations, the authors constructed new YKL075C deletion strains to verify that the phenotypes observed were linked to the ORF. However, this it not a sufficient protocol to test this as the deletion could have polar effects on both downstream and upstream genes. This is particularly important here because another ORF (YKL076C) overlaps with the C-terminal region of YKL075C. A standard complementation test should be performed (providing the ORF in trans) to make sure the phenotypes are linked to the YKL075C ORF.

We inserted the *YKL075C* promoter and gene into the yeast genome at the *HO* locus of *ykl075c* Δ cells and found that expression of *YKL075C* in *ykl075c* Δ cells restores actin cable abundance to levels observed in wild-type cells. Thus, we obtained additional evidence that the phenotypes observed in the *ykl075c* Δ mutant are due to loss of the *YKL075C* gene. The new data has been added to the revised manuscript (Extended Data Fig. 2a,b).

2. The YKL075C ORF was identified here by virtue of the strain carrying this deletion being less sensitive to Lat-A. Did the authors test if this could be due also to effects on Lat-A import or export. This seems relevant to test since the phenotypes linked previously to a YKL075C deletion includes effects on 'drug export' and 'responses to drugs'. I do believe the authors have good reasons to conclude that the ORF affects actin cables (in a Lat-A independent manner) since they observe affects without Lat-A but may want to comment on the possibility that the ORF also effects Lat-A uptake/export.

A similar argument could be applied to the differential effects of Lat-A in young and old cells – e.g. are old cells displaying defects in e.g. drug export?

As stated by the reviewer, there is good reason to believe that the changes in actin abundance observed upon deletion of *YKL075C* are not due to effects on Lat-A import/export. That said, it is possible that Lat-A uptake or export declines with age. Moreover, some of the hits identified in our screen may have an impact on Lat-A uptake or export. Specifically, cases in which gene deletions result in reduced sensitivity to the growth-inhibiting effects of Lat-A, but have no effect on actin cable abundance in the absence of Lat-A treatment may well be due to effect on uptake or export of the drug. Indeed, as noted by the reviewer, this phenotype was observed in mutations that impact on membrane trafficking. We modified the text to comment on possible effects on Lat-A import/export in hits identified in the screen.

3. It is somewhat difficult to follow the authors' conclusions concerning Tor: One of the conclusions states that the results indicate that Aan1p function is TORC1-independent. However, what is measured is the recovery of mutant cells after the removal of rapamycin from the medium and the authors found that cells completely devoid of *Tor1* recovers slowly (as expected) whereas *aan1* Δ cells did not. While this shows that *aan1* Δ cells do not show the same defect as TOR1-deleted cells, it does not necessarily demonstrate that the effects of deleting *AAN1* is completely independent on TORC1. For example, does the *aan1* Δ have the same effect in mutants with constitutive Tor activity? Also, the authors state that 'we find that TORC1 activity is not affected by deletion of *AAN1*'. This is not measured in the paper. To do so, the authors could use the standard methods of measuring RPS6 phosphorylation or a shift of HA-tagged Sch9. More experiments or a rephrasing of the text might be needed here.

In response to concerns raised by Reviewers 1 and 2, we analyzed the phosphorylation of a direct downstream target of TORC1, ribosome protein S6 (Rps6p). We found that deletion of *AAN1* has no effect on Rps6p phosphorylation and added the new data to the revised manuscript (Fig. 3i,j). These findings, together with the results obtained in the recovery from rapamycin withdrawal assay, indicate that Aan1p is independent of TORC1.

4. To check the localization of the ORF protein, the authors tagged the gene with 13 copies of the Myc epitope and, using immunofluorescence, found that Ykl075cp localized to punctate cytoplasmic structures that do not co-localize with actin cables. They further show that the tag had no effects on

the cell growth rate. However, to test if it is functional, this analysis needs to be complemented with analysis of whether cells carrying the tagged version, when being the sole source of Ykl075cp in the cell, is not displaying any of the phenotypes displayed by cells harboring the deleted ORF. Also, it is difficult to ascertain that the tag is not causing effects on localization and foci formation without further controls. I think the authors might want to consider removing this data set in the paper as it does not contribute much to the otherwise interesting story and raises several questions.

We expressed untagged and Myc-tagged *YKL075C* under control of the *YKL075C* promoter in *ykl075cΔ* cells. We found that Myc-tagged Ykl075cp is expressed at levels similar to that of endogenous Ykl075cp and that expression of Myc-tagged or untagged *YKL075C* in *ykl075cΔ* cells fully rescue changes in actin cable abundance to wild-type levels. This data, which has been added to the revised manuscript (Extended Fig. 3b,c), strengthens our evidence that localization of Ykl075p is not an artifact of the Myc tag.

5. As it has been shown previously that increasing actin dynamics, either by specific conditional actin alleles or by removing the actin-bundling protein Scp1p, increases lifespan, accompanied by affects on the polarization of the mitochondrial membrane and ROS production (Gourlay et al. JBC, 2004), it would be interesting to hear the authors view on whether Aan1 and Scp1 may act in the same pathway of lifespan control and whether deleting these two genes are additive or not.

Scp1p and Aan1p affect actin stability, mitochondria and lifespan, but appear do so by different mechanisms. Scp1p binds to actin and localizes to actin patches. Aan1p impacts on actin cables but does not have direct interactions with actin cables. Moreover, these proteins have different effects on actin: Scp1p stabilizes actin patches, whereas Aan1p destabilizes actin cables. Finally, although both proteins affect mitochondria, ROS and lifespan, they do so during different stages of the yeast life cycle. Scp1p affects chronological lifespan (survival of stationary phase, non-dividing yeast). In contrast, Aan1p affects replicative lifespan (survival and replication competence of rapidly growing yeast cells). These observations underscore the complexity of the actin cytoskeleton and its impact on mitochondria and lifespan.

Comment: There is some data in the Rizzolo et al., paper (2017, Cell Reports) that might be helpful to the further analysis by the authors of YKL075C as the ORF has been linked, by genetic/physical interaction studies to the 'Chaperone Cellular Network', and especially linked to the GO function 'Chromatin and Nucleic acid'. (Figure 3 and Figure S6 in the paper). This paper reports both genetic and physical interaction in terms of supercomplexes of chaperones and co-chaperones. They also note the YKL07C forms foci in stationary phase and has correlation to *cdc37*. the previous report show there are negative genetic interactions with *cdc37* and *act1* among other.

Thank you for this information. We will take these findings into account in future studies on Aan1p.

Reviewer #2 (Remarks to the Author):

This is a great paper where the authors identify a novel gene target for regulation of actin cable stability in yeast, and demonstrate that deletion of this regulator actin cable stability is increased for ageing cells and this increases viability of these cells. The authors identify that the new gene target may be linked to branch chain amino acid (BCAA) metabolism, and find that a similar effect can be found by either deleting a biosynthetic gene associated with BCAA biosynthesis or by reducing leucine levels in the medium.

The paper presents some very solid results and the findings are surely of sufficient general interest to merit publication in Nature Com, but the story needs a little more additional work in order to be sufficiently conclusive.

The authors basically find that deletion of *AAN1* (the new target gene identified) gives the same effect as deleting of *BAT1*, that is engaged in BCAA biosynthesis. They also found that BCAA levels in both strains decreases and this points to a role of BCAA, which is further confirmed by an effect of eliminating leucine from the medium. The authors do not see any effect on TORC1. This opens up for the following questions that should be addressed:

1) Could Aan1p be working downstream of TORC1? Could maybe be tested by phospho-proteomics of Aan1p with or without leucine in the medium.

In response to concerns raised by Reviewers 1 and 2, we analyzed the phosphorylation of a direct downstream target of TORC1, ribosome protein S6 (Rps6p). We found that deletion of *AAN1* has no effect on Rps6p phosphorylation and added the new data to the revised manuscript (Fig. 3i,j). These findings, together with the results obtained in the recovery from rapamycin withdrawal assay, indicate that Aan1p is independent of TORC1.

2) Is there a link to energy metabolism? BCAA can serve as an important energy source, and BCAA catabolism plays an important role in some cancer cells.

BCAAs are an important energy source for cancer cells. However, we find that depletion of BCAAs promotes actin cable abundance. Specifically, deletion of *AAN1* results in a decrease in BCAA biosynthesis and a decrease in BCAA levels, which in turn result in an increase in actin cable abundance and stability. Moreover, depletion of leucine increases actin cable abundance, and leucine supplementation has the opposite effect. Therefore, it is unlikely that BCAAs are serving as an energy source for cytoskeleton remodeling. Instead, we favor the model that Aan1p functions in a nutrient sensing pathway that responds to leucine levels, and revised the discussion to describe this possible function of Aan1p.

3) Could the authors express AAN1 heterologously to evaluate if it a DNA binding protein, i.e. is it a transcription factor.

We do not detect localization of Aan1p to the nucleus or any domains in Aan1p that suggest that it is a DNA binding protein. Therefore, although deletion of *AAN1* results in changes in gene expression, available evidence indicate that it is not a transcription factor.

Maybe not all experiments are needed, but some more hints towards a possible function of Aan1p would significantly increase the value of the paper.

We suspect that Aan1p functions in a nutrient sensing pathway that impacts on actin cytoskeletal, mitochondria and lifespan. Our studies indicate that deletion of *AAN1* does not activate TORC1. However, it is possible that Aan1p affects the General Amino Acid Control pathway (GAAC), a signaling pathway that regulates yeast cell growth, metabolism and translation in response to amino acid limitations. GAAC affects expression of amino acid biosynthesis genes, including *BAT1*, and aminoacyl-tRNA synthetase. Moreover, the actin cytoskeleton has been linked to GAAC function in translational regulation. We included a comment on a possible links between Aan1p and GAAC in the revised manuscript, as suggested.

Reviewer #3 (Remarks to the Author):

Sing et al. report that the stability of actin cables apparently declines with age in *Saccharomyces cerevisiae*. By screening a yeast deletion library, the authors identified a previously uncharacterized gene (YKL075C) as being an important factor required for proper actin cable stability and abundance, mitochondrial function and branched-chain amino acid (BCAA) metabolism. Based on the phenotypic analysis, they name YKL075C *AAN1* (actin, aging and nutrient regulatory protein 1). *AAN1* is not an essential gene, but strains carrying null alleles exhibit enhanced mitochondrial function, altered leucine catabolism and increased replicative lifespan.

Overall, the experimental execution and phenotypic analyses are competently done, and manuscript is very well written. However, the major conclusions are not placed in context with previous studies showing the importance of leucine in longevity studies. Furthermore, the results from the screen leading to the identification of *AAN1* are merely cursorily discussed. The inactivation of *ANN1* was one of 18 null mutations that resulted in resistance to low levels of latrunculin A, and one of 15 that correlated with increased actin cable stability. The lack of discussion regarding the other null mutations makes the focus and interpretation on Ann1 function harder to understand, the overall

context is missing. The authors applied transcriptomic analysis to understand Ann1 function, which led to the insight that leucine metabolism has a link to actin cable stability. Although, this is an important step forward, in the end, the manuscript does not provide a framework to place this finding in any mechanistic perspective. Consequently, the manuscript provides descriptive information that represents novel information of a little studied gene, but ultimately provides limited novel insight to advance the understanding of the coupling between BCAA metabolism, actin cable stability and longevity.

The authors may want to consider the following points:

Major comments:

1. to place the results regarding leucine in perspective and to strengthen their conclusion the authors could acknowledge previous work including: Aris et al. (2013) Autophagy and leucine promote chronological longevity and respiration proficiency during calorie restriction in yeast. *Experimental Gerontology* 48:10, 1107-1119; Maruyama et al (2016) Availability of Amino Acids Extends Chronological Lifespan by Suppressing Hyper-Acidification of the Environment in *Saccharomyces cerevisiae* <https://doi.org/10.1371/journal.pone.0151894>

We included a description of work by Aris et al. 2013, and Maruyama et al., 2016 in the revised manuscript.

2. the latrunculin A screen proved to be very informative. A brief discussion regarding the other genes identified in the screen would add significantly, perhaps organized with potential function. For example, Far10 and Bro1 did not affect number of actin cables but exhibited resistance to Lat-A. Far10 is a paralog of VPS64, Bro1 is involved in protein turnover – Rsp5 – MVB pathway. Several other genes may affect the MVB pathway, including Rgp1, Swf1, Roq1 (YJL144W - why not use gene name?). Could this provide clues that BCAA uptake is affected?

We expanded our analysis and discussion of the other hits identified in the screen in the revised manuscript. First, we carried out SGD GO SLIM analysis of the 17 hits in known genes, which revealed a role for 2 or more of the hits in processes including response to chemical stressors, cell cycle, DNA recombination, regulation or repair, amino acid metabolism, translation, protein targeting and endosomal transport (Extended Data Fig. 1e). Endosomal transport impacts on the amino transport and on the organelle (the vacuole) that stores and contributes to amino acid sensing. The finding that other hits function in amino acid metabolism and endocytosis raises the possibility that these hits may contribute to Aan1p function in modulation of the actin cytoskeleton in response to BCAAs. Moreover, since 2 hits (*BRO1* or *FAR10*) that function in membrane trafficking result in reduced sensitivity to the growth-inhibition by Lat-A, but have no effect on actin cable abundance, these deletions may be due to effects on uptake or export of the drug. We modified the text to include a discussion of other hits identified in the screen.

Also, several of the genes have known mitochondrial function, e.g., SNO4, IXR1, IDH1 and AFG1. How does the deletion of these genes square with the idea that deletion of ANN1 leads to increased mitochondrial function?

Although deletion of *AAN1* promotes mitochondria function and lifespan, other hits identified in our screen may promote actin cables by *AAN1*-independent mechanisms. For example, our previous studies indicate actin cables serve not only as tracks for movement of mitochondria from mother cells to buds, but also for preferential inheritance of fitter mitochondria by buds. Therefore, it is possible that defects in mitochondrial function may lead to compensatory increases in actin cables.

Also, the finding that ARO1 is potentially interesting, or? Finally, YHR022C is an ORF of unknown function, why focus on YKL075C?

We focused on *YKL075C* because it is an uncharacterized open reading frame and because newly generated *ykl075cΔ* strains exhibited the same phenotype observed in the initial screen: reduced sensitivity to the growth-inhibited effects of low-level Lat-A treatment and increased actin cable abundance.

3. It is difficult to assess what strains were used in the figures. What data stems from leu2 auxotrophs and what data is derived from LEU2 prototrophs?

The identification and initial characterization of *AAN1* were carried out only in auxotrophs. So, auxotrophs were used for the screen (Fig. 1e-f; Extended Fig. 1d-e), growth curves on newly synthesized *yki075cΔ* strains (Fig. 2a), replicative lifespan measurements (Fig. 2f; Extended Fig. 2f), RNA seq analysis (Fig. 3b-c; Supplemental Table 1) and qPCR validation (Fig. 3d-e). All studies on the role of BCAAs in modulating actin cables (Fig. 3f-j, Fig. 4 and Extended Data Fig. 4) were carried out in prototrophs. All other studies were conducted in auxotrophs and prototrophs, and the results obtained were identical in both strain types.

4. The punctate localization of the myc-tagged Ann1 is intriguing. Does Aan1 co-localize with Golgi or endosomal markers?

We added images of the localization of Aan1p and the actin cytoskeleton to the revised manuscript (Fig. 3a). The new data indicate that punctate Aan1p containing structures do not co-localize with actin patches. Since actin patches are endosomes coated with F-actin, there is no physical link between Aan1p and endosomes. Equally important, the new data provide documentation that Aan1p puncta do not co-localize with actin cables. These findings indicate that Aan1p affects actin cable stability through indirect effects on those structures.

5. decreases BCAA biosynthesis appears to be a key feature, and the simultaneous enhanced expression of BCAA catabolism is interesting. The increased levels of BAT2 observed in ann1 null mutants did not come up in the transcriptomics analysis, is there a reason for this?

The RNAseq analysis did reveal an increase in *BAT2* transcripts. However, the increase was just beyond the cut-off for significance ($p=0.053$). We used qPCR analysis to further characterize the links between BCAAs and *AAN1* and observed an increase in *BAT2* transcripts that are statistically significant ($p=0.0033$).

6. the authors write that Ann1 regulates actin cables, key nutrients, and lifespan in yeast. The term regulate is a strong word – the data merely show that AAN1 is required for proper actin cable thickness, BCAA metabolism and cell longevity – where is the data to substantiate regulation?

This concern is reasonable. We modified the text accordingly.

7. SD-leu represents a very intense leucine starvation conditions for leu2 strains. Is there any indication of a stress response may increase/stabilize actin cables? Added leucine suppresses the actin stabilisation, which suggests that leucine limitation is important.

To avoid leucine starvation, we conducted leucine depletion or supplement studies for short periods of time (20 min) in prototrophic strains that have biosynthetic activity for leucine and all amino acids. Moreover, there was no indication that the conditions used resulted in cellular stress. For example, complete destabilization of the actin cytoskeleton, a phenotype produced by acute stressors (e.g heat shock or shear stress), was not apparent in our studies. Indeed, we found that short-term leucine depletion promotes cable abundance and stability. Moreover, although leucine supplementation reduces actin cable stability and polarity, it does not produce widespread loss of the actin cytoskeleton. Therefore, the short-term leucine depletion or supplementation conditions used for our studies do not appear to induce severe cellular stress. We revised the manuscript to discuss this important issue.

8. Does D-leucine suppress the latrunculin A affect on actin cables? The thought being that leucine may have an allosteric role in this activity.

This experiment is on the list for future *AAN1* studies. To address the issue of amino acid specificity, we tested whether the observed cytoskeletal phenotypes are produced by modulation of an amino acid that is not a BCAA. We found that depletion of lysine has no obvious effect on actin cable abundance. This new data has been added to the revised manuscript (Extended Data Fig. 4c,d).

Minor comments:

1. Healthspan? Quality of life during aging? Is this an accepted term for yeast? How to assess "quality" of life in yeast?

Healthspan is used to define quality of life in yeast. Early studies revealed that the double time (mean generation time) of yeast increases with replicative age in yeast (Mortimer and Johnston, 1959). Therefore, mean generation time is used as a readout for healthspan in yeast.

2. Fig 1e – the color scheme used makes it difficult to follow the growth of the strains. These curves appear to be generated using microtiter plate readers with short pathlengths for OD measurements – hence the low ODs of saturated cultures. It would be good to state in the methods how the curves were generated. By contrast, the OD measurements in Fig. 2a end up at higher ODs, likely to have determined using standard spectroscopic measurements using a 1 cm pathlength cuvette. Also, the growth curves are clearly biphasic, the mutants start to grow and then cease, seemingly to recover after 30 hr. What is going on here? Do the authors have thoughts regarding this?

Good call! We did use a plate reader and conventional spectrophotometer to assess growth rates in Fig. 1e and 2a, respectively. The methods section has been revised to more accurately describe the methods used.

Lat-A treatment results in cell clumping, which is not disrupted by the shaker in our plate reader but is disrupted by more vigorous mixing in manual OD measurements. There is a lot of noise in growth curves of *yki075c*Δ strains propagated +Lat-A measured in our plate reader that could be interpreted as biphasic (Fig. 1e). However, the growth of *yki075c* strains obtained manually is not biphasic (Fig. 2a). Therefore, we suspect that the apparent biphasic behavior is a consequence of cell clumping in the plate reader.

Since it is difficult to follow the growth curves in Fig. 1e, we added data showing the growth rate of each hit obtained in the screen (Extended Fig. 1d).

3. Fig. 1e,f - in Fig.1f Lsm6 is used but in Fig. 1e. ism6 is used as strain label – ism6 does not exist in SGD, probably a mistake in the label in Fig. 1e?

Thank you for this comment. We corrected the strain name in Fig. 1e.

REVIEWER COMMENTS

Reviewer #1 (Remarks to the Author):

I think the authors have responded satisfactorily to all me concerns and questions.

Reviewer #2 (Remarks to the Author):

I am satisfied with the revision and I do not have any further comments. This is a great paper

Reviewer #3 (Remarks to the Author):

The authors have substantially improved the manuscript and have addressed many of the concerns raised in the initial round of review. In making adjustments, the authors included references to previous work and clearly demonstrate that whatever Ann1 does, it does so independently of TorC1. Clarification and new data regarding the intracellular localization of Aan1-13myc is also provided, demonstrating that the Ann1 puncta do not co-localize with actin cables and thus apparently are not endosomal. This new data is enlightening, however mechanistic insights are lacking as to how loss of *ANN1* leads to a decreased leucine (BCAA) biosynthetic capacity, increased actin cable stability and longevity.

The authors posit that Ann1 is involved with a nutrient sensing pathway and even suggest that this sensing pathway impinges on general amino acid control (GAAC), which is intriguing. Apparently, the hypothesis is that Ann1 activity is required to maintain BCAA biosynthesis and conversely negatively regulates *BAT2* in logarithmically growing cells; inactivation of *ANN1* manifests repressed biosynthetic genes (e.g., *BAT1* and *ILV2*) and derepression of *BAT2*. However, the authors have not directly tested this hypothesis or discussed this possibility with respect to their transcriptomic analysis; how consistent is this with their findings, is there a clear and definate GAAC signature? Also, the possibility that *ann1*Δ mutants exhibit altered translational initiation could be tested by monitoring Gcn4 expression or stability. How does the observed intracellular distribution (puncta) of Ann1 fit this model? Ann1 colocalization with other subcellular structures/organelles could be examined, perhaps Golgi or vacuole, using subcellular fractionation or microscopically using known marker proteins.

Remaining Concerns: Reviewer 3

Reviewer 3 suggested that we test whether the General Amino Acid Control (GAAC) pathway plays a role in Aan1p function in leucine (BCAA) biosynthesis, actin cable stability and lifespan. We developed the approach outlined below, which includes specific experiments described by Reviewer 3, to address this concern.

Is GAAC a downstream target of Aan1p?

- a. Does deletion of *AAN1* affect the levels of genes involved in the GAAC pathway?
Analysis of the steady state levels and turnover of *GCN4*, a key GAAC regulator in WT and *aan1Δ* cells
- b. Is the pattern of gene expression observed in response to deletion of *AAN1* similar to that observed upon activation of GAAC and if so, is Aan1p required for GAAC function in regulation of gene expression?
RNA seq analysis of in WT, *aan1Δ*, *gcn4Δ*, and *aan1Δ gcn4Δ* cells
- c. Is GAAC required for changes in BCAA levels observed upon deletion of *AAN1*?
Analysis of BCAA levels in WT, *aan1Δ*, *gcn4Δ*, and *aan1Δ gcn4Δ* cells

Does GAAC regulate actin cable abundance, stability and function and if so, does Aan1p modulate GAAC function in actin cable regulation? Note: Although GAAC has known effects on Aan1p targets (BCAA levels and lifespan), there is no evidence for a role for GAAC in regulation of the actin cytoskeleton.

- Analysis of actin cable abundance in WT, *aan1Δ*, *gcn4Δ*, and *aan1Δ gcn4Δ* cells
- Analysis of the sensitivity to actin cables to LatA treatment in WT, *aan1Δ*, *gcn4Δ*, and *aan1Δ gcn4Δ* cells
- Analysis of actin cable function (e.g. mitochondrial distribution, motility and/or redox state) in WT, *aan1Δ*, *gcn4Δ*, and *aan1Δ gcn4Δ* cells

Is Aan1p and its function in regulation of actin cables required for Gcn4p control of lifespan?

- Analysis of RLS in WT, *aan1Δ*, *gcn4Δ*, and *aan1Δ gcn4Δ* cells
- Analysis of actin cable abundance as a function of replicative age in WT, *aan1Δ*, *gcn4Δ*, and *aan1Δ gcn4Δ* cells
- Analysis of the sensitivity to actin cables to LatA treatment as a function of replicative age in WT, *aan1Δ*, *gcn4Δ*, and *aan1Δ gcn4Δ* cells
- Analysis of actin cable function (e.g. mitochondrial distribution, motility and redox state) as a function of replicative age in WT, *aan1Δ*, *gcn4Δ*, and *aan1Δ gcn4Δ* cells
- Analysis of the effect of actin cable stabilization on RLS in in WT, *aan1Δ*, *gcn4Δ*, and *aan1Δ gcn4Δ* cells
- Analysis of the effect of increasing actin cable abundance on RLS in in WT, *aan1Δ*, *gcn4Δ*, and *aan1Δ gcn4Δ* cells

In the manuscript under consideration, we carried out a genome-wide screen for genes that regulate actin cable stability. Our studies revealed a novel role for a previously undescribed open reading frame (YKL075c aka *AAN1*) in control of the actin cable stability and abundance as well as mitochondrial quality and lifespan. Our finding that stabilization of the actin cables results in lifespan extension in yeast, as in *C. elegans*, supports a conserved function for actin cytoskeletal stability in the aging process. In addition, we identified the mechanism underlying Aan1p function in these processes. Specifically, we identified a novel role for BCAA and more specifically for leucine in control of the actin cytoskeleton and mitochondrial quality. Based on the response of all three reviewers, there is no question regarding the novelty, mechanistic insight or impact of our findings. Reviewer 3's suggestion that we test whether GAAC is critical for Aan1p control of BCAA, actin organization and lifespan is interesting and insightful. However, this line of investigation is beyond the scope of this study.